# Development and Characterization of Sustainable Composites from Bacterial Polyester Poly(3-Hydroxybutyrate-*co*-3-hydroxyhexanoate) and Almond Shell Flour by Reactive Extrusion with Oligomers of Lactic Acid

**DOI:** 10.3390/polym12051097

**Published:** 2020-05-11

**Authors:** Juan Ivorra-Martinez, Jose Manuel-Mañogil, Teodomiro Boronat, Lourdes Sanchez-Nacher, Rafael Balart, Luis Quiles-Carrillo

**Affiliations:** Technological Institute of Materials (ITM), Universitat Politècnica de València (UPV), Plaza Ferrándiz y Carbonell 1, 03801 Alcoy, Spain; jomaoam@epsa.upv.es (J.M.-M.); tboronat@dimm.upv.es (T.B.); lsanchez@mcm.upv.es (L.S.-N.); rbalart@mcm.upv.es (R.B.); luiquic1@epsa.upv.es (L.Q.-C.)

**Keywords:** PHBH, almond shell flour, mechanical properties, thermal characterization, WPCs

## Abstract

Eco-efficient Wood Plastic Composites (WPCs) have been obtained using poly(hydroxybutyrate-co-hexanoate) (PHBH) as the polymer matrix, and almond shell flour (ASF), a by-product from the agro-food industry, as filler/reinforcement. These WPCs were prepared with different amounts of lignocellulosic fillers (wt %), namely 10, 20 and 30. The mechanical characterization of these WPCs showed an important increase in their stiffness with increasing the wt % ASF content. In addition, lower tensile strength and impact strength were obtained. The field emission scanning electron microscopy (FESEM) study revealed the lack of continuity and poor adhesion among the PHBH-ASF interface. Even with the only addition of 10 wt % ASF, these green composites become highly brittle. Nevertheless, for real applications, the WPC with 30 wt % ASF is the most attracting material since it contributes to lowering the overall cost of the WPC and can be manufactured by injection moulding, but its properties are really compromised due to the lack of compatibility between the hydrophobic PHBH matrix and the hydrophilic lignocellulosic filler. To minimize this phenomenon, 10 and 20 *phr* (weight parts of OLA-Oligomeric Lactic Acid per one hundred weight parts of PHBH) were added to PHBH/ASF (30 wt % ASF) composites. Differential scanning calorimetry (DSC) suggested poor plasticization effect of OLA on PHBH-ASF composites. Nevertheless, the most important property OLA can provide to PHBH/ASF composites is somewhat compatibilization since some mechanical ductile properties are improved with OLA addition. The study by thermomechanical analysis (TMA), confirmed the increase of the coefficient of linear thermal expansion (CLTE) with increasing OLA content. The dynamic mechanical characterization (DTMA), revealed higher storage modulus, E’, with increasing ASF. Moreover, DTMA results confirmed poor plasticization of OLA on PHBH-ASF (30 wt % ASF) composites, but interesting compatibilization effects.

## 1. Introduction

The current problem related to the negative environmental impact of large volumes of wastes [1] in a consumer society has promoted a significant awareness and sensitiveness about this problem. Some governments are facing this through legislation that protects environment and minimizes the harmful impact on nature. This, in part, has led to the extensive development of new eco-efficient materials, from the point of view of their renewable origin, low carbon footprint, possibility of composting, biodegradability, and so on [2]. An interesting family of these new eco-efficient materials are the so-called Wood Plastic Composites, WPC. These composites consist on a polymeric matrix in which wood (or whatever lignocellulosic subproduct of the food industry or agroforestry) particles (from 10 to 60 wt %, depending on the manufacturing process) are embedded, leading to an appearance and surface finishing similar to natural wood. As many times, the lignocellulosic fillers are by-products from other sectors, they are cheap and do not increase the cost of the WPC; in addition, they come from natural resources and, subsequently, they represent a sustainable source for use in new and environmentally friendly materials [3,4,5].

These WPCs are already replacing the use of traditional woods in some applications, which is an important protection of forest resources. WPCs formulations have been optimized in sectors as important as automotive, outdoor furniture, interior design, railings, floors, coatings, decks, fences, pergolas, decking, and so on [3,4,5,6,7,8].

The fact that they are composed of a polymeric matrix, gives them better behaviour against water or in humid environments. According to Singh et al., WPCs have gained a significant share of the consumer market, becoming the fastest growing segment of the plastics industry [4]. Within the wide range of possibilities that these eco-efficient materials offer as substitutes for wood, those that use thermoplastic polymers as the matrix are of particular interest, precisely because of the ease and versatility of manufacturing processes. Poly(ethylene) (PE), poly(styrene) (PS), poly(vinyl chloride) (PVC) and poly(propylene) (PP) are some of the most widely used polymers in WPCs. Nevertheless, these polymer matrices are petroleum-derived polymers.

Due to the need to protect the environment, the possibility of using biopolymers as matrices in WPCs is currently being studied. The use of a biodegradable thermoplastic polymer (actually, a compostable polymer that disintegrates in controlled compost soil) from natural resources, together with a natural lignocellulosic filler from industrial wastes or by-products, allows the obtaining of totally biodegradable and eco-efficient WPCs [6]. These new green composites represent the new generation of biobased, sustainable, low environmental impact WPCs. Nowadays, there are already a large number of natural biopolymers on a commercial level, among which three main families stand out. The first one, consists on polymers from biomass which include polysaccharides such as starch (and starch-derived polymers such as polylactide), cellulose, chitosan, chitin, and proteins such as casein, keratin, collagen, and so on. The second group includes conventional polymers such as poly(ethylene), poly(urethanes), poly(amides), that are partially or fully obtained from natural resources but they show identical (or very similar) properties to their petroleum-derived counterparts. Finally, a new family of very promising polymers is that of bacterial polyesters which are generally referred to as polyhydroxyalkanoates PHAs. PHAs include more than 300 different polyesters and copolymers such as poly(3-hydroxybutyrate) (P3HB or just PHB), poly(3-hydroxybutyrate-co-3-hydroxyvalerate) (PHBV), among others [5,9].

One of the most interesting biopolymers, obtained by bacterial fermentation, is poly(3-hydroxybutyrate-*co*-3-hydroxyhexanoate), PHBH. This copolymer is obtained by incorporating into the polyhydroxybutyrate chain, PHB, 3-hydroxyhexanoate units with medium-length side groups, [P(3HB-co-3HH)]. Mahmood and Corre identify a structure formed by branches of short 3HH units, on the main 3HB chain, thus reducing regularity. Yang and Liao compare the formation of these units by dielectric spectroscopy and melt viscosity [10,11]. Moreover, the addition of the 3HH units extends the temperature range for processing of this copolymer, but the storage modulus and the strength is reduced [12,13,14,15]. Watanabe and Oyama, synthesized PHBH from cheap natural resources such as coconut oil, biomass, beet, sugar cane, molasses and vegetable oils [16,17]. These characteristics allow it to be used as a substitute for traditional petroleum-derived polymers in some applications, such as disposable plastic bags, food packaging, catering, agricultural mulch film, and so on [18].

The main objective of this work is to obtain fully biobased WPCs. For this purpose, PHBH was chosen as the thermoplastic matrix; this matrix was reinforced with almond shell flour (ASF). The almond shell flour, ASF, is a waste of the agro-food industry. It is very cheap, fully biobased and biodegradable. By incorporating ASF into the PHBH matrix it gives a wood-like appearance. In this work, the effect of the amount of ASF on the mechanical, thermal, thermomechanical and water absorption properties of PHBH-ASF composites is investigated. In addition, the optimization of the behaviour of these composites is used by the addition of an oligomer of lactic acid (OLA), to provide some plasticization and to increase toughness. Due to the lack of compatibility between the different elements, reactive extrusion (REX) has been proposed as a strategy to improve the properties of the mixtures. This process will improve the chemical bonding of the biopolymer chains to the surface of the lignocellulosic fillers by the action of reactive molecules with at least two functional sites.

## 2. Experimental Section

### 2.1. Materials

The PHBH commercial grade (ErcrosBio^®^ PH 110) used in this study was supplied in pellet by Ercros S.A. (Barcelona, Spain). This polymer has a density of 1.2 g cm^−3^ and a melt flow index (MFI) of 1 (g/10 min^−1^) measured at 160 °C. Even with this low MFI, this is suitable for injection moulding as it has very low melt strength, so requires an appropriate temperature profile for extrusion and injection moulding. Almond shell powder/flour (ASF) was purchased from Jesol Materias Primas (Valencia, Spain). This powder was sieved in a vibrational sieve RP09 CISA^®^ (Barcelona, Spain) to obtain a maximum particle size of 150 µm. Figure 1 shows the irregular particle size of ASF with average size below 150 μm (the average size is 75 μm). As plasticizer/impact modifier, an oligomer of lactic acid (OLA), commercial grade Glyplast OLA 8 was kindly provided by Condensia Química S.A. (Barcelona, Spain). Glyplast OLA 8 is a liquid polyester (with an ester content above 99%) with a viscosity of 22.5 mPa s measured at 100 °C. Its density is 1.11 g cm^−3^; it has a maximum acid index of 1.5 mg KOH g^−1^ and a maximum moisture content of 0.1%.

### 2.2. Manufacturing of PHBH-ASF Composites

Before further processing of composites, PHBH pellets and almond shell flour were dried for 6 h at 80 °C, in a dehumidifier model MDEO, supplied by Industrial Marsé (Barcelona, Spain). Then, different amounts (see Table 1) of PHBH, ASF (in wt %) and OLA (in *phr*–weight parts of OLA per one hundred weight parts of PHBH) were mechanically pre-mixed in a zipper bag to obtain pre-homogenization. These six materials were then extruded in a twin-screw corotating extruder from DUPRA S.L. (Alicante, Spain). The four temperature barrels were programmed to the following temperature program: 110 °C (hopper), 120 °C, 130 °C and 140 °C (extrusion die) and the screw speed was maintained in the 20–25 rpm range. The extruded material was cooled down to room temperature and then, pelletized for further processing by injection moulding. The injection moulding process was carried out in a Sprinter 11 injection machine from Erinca S.L. (Barcelona, Spain) to obtain standard samples for further characterization. As PHBH has low melt strength, it needs some particular processing conditions. The injection temperature profile was set to 150 °C (hopper), 140 °C, 130 °C and 120 °C (nozzle). In addition, it requires a tempered mould at 60 °C. The filling and cooling times were set to 1 s and 20 s, respectively. It is well known that bacterial polyesters undergo secondary crystallization or recrystallization with time (sometimes designed as physical aging since this leads to an embrittlement), especially at temperatures above *T_g_*. Recrystallization rate is directly related to temperature; therefore, samples have been subjected to a recrystallization process at 25 °C for 15 days since it has been reported that almost all recrystallization takes place after two weeks from the processing [17,19,20]. To avoid potential hydrolysis of the polyester surface, samples were stored in a vacuum desiccator with constant moisture.

### 2.3. Mechanical Properties of PHBH-ASF/OLA Composites

The mechanical characterization of PHBH-ASF/OLA composites was carried out by means of tensile tests according to ISO 527-2:2012 in a universal testing machine, model ELIB-50 from Ibertest (Madrid, Spain). A 5 kN load cell was used and the crosshead rate was set to 10 mm min^−1^. The standardized specimens corresponded to the designation A12 from ISO 20753:2018. Impact resistance was quantified by means of a Charpy test, with a 1-J pendulum from Metrotec S.A. (San Sebastian, Spain), on specimens with a standardized “V” notch, according to ISO 179-1:2010. In addition, the hardness of PHBH-ASF/OLA composites, was obtained using a Shore-D hardness tester model 673-D from J. Bot Instruments, S.A. (Barcelona, Spain) according to ISO 868:2003. All mechanical tests were performed on 5 specimens of each composition.

### 2.4. Morphology of PHBH-ASF/OLA Composites

The morphology study of the impact fractured specimens from impact tests was carried out by field emission scanning electron microscopy (FESEM) in a ZEISS ULTRA 55 microscope from Oxford Instruments (Abingdon, Oxfordshire, UK). The accelerating voltage was 2 kV. Prior to this analysis, the samples were metallized with platinum in a sputtering metallizer EMITECH mod. SC7620 from Quorum Technologies Ltd. (East Sussex, UK).

### 2.5. Thermal Characterization of PHBH-ASF/OLA Composites

The thermal characterization of PHBH-ASF/OLA composites, by means of differential scanning calorimetry (DSC), was performed in a TA Instruments calorimeter mod. Q2000 (New Castle, DE, USA). For the thermal study, a dynamic temperature cycle was scheduled with the following sequence: 1st cycle: −50 °C to 200 °C at a constant heating rate of 10 °C min^−1^, 2nd cycle: 200 °C to −50 °C at a constant cooling rate of 10 °C min^−1^; this step was scheduled to remove the thermal history. Finally, a 3rd cycle from −50 °C up to 200 °C at 10 °C min^−1^ was programmed. The DSC analysis was performed in an inert nitrogen atmosphere with a flow rate of 50 mL min^−1^, with samples between (5–10 mg), in standard 40 μL aluminium crucibles. The degree of crystallinity (*X_c_*) was calculated by using Equation (1) where $\Delta H_{m}$ and $\Delta H_{cc}$ (J g^−1^) are melt enthalpy and cold crystallization enthalpy respectively. $\Delta H_{m}^{0}$ (J g^−1^) is the theoretical value that corresponds to fully crystalline PHBH; this was taken as 146 (J g^−1^) as reported in Reference [14]. Finally, w is the fraction weight of PHBH.
(1)$$X_{c}=\left[\frac{\Delta H_{m}-\Delta H_{cc}}{\Delta H_{m}^{0}\times w}\right]\times 100$$

Thermogravimetric analysis (TGA) was carried out in a Mettler-Toledo TGA/SDTA 851 thermobalance (Schwerzenbach, Switzerland). Samples consisted on small pieces with a total weight of 5–7 mg. These samples were placed in standard alumina pans (70 μL), and then subjected to a heating ramp from 30 to 700 °C at a constant heating rate of 20 °C min^−1^ in nitrogen atmosphere. All the thermal tests were done in triplicate.

### 2.6. Thermomechanical Characterization of PHBH-ASF/OLA Composites

The dynamic-mechanical-thermal analysis, DMTA, was done in a Mettler-Toledo dynamic analyzer (Columbus, OH, USA), on rectangular samples of 40 × 10 × 4 mm^3^. Heating was programmed from −70 °C to 70 °C at a constant rate of 2 °C min^−1^; samples were subjected to a single cantilever test in dynamic conditions with a maximum deflection of 10 µm and a frequency of 1 Hz. The coefficient of linear thermal expansion (CLTE) of the PHBH-ASF/OLA composites was determined using a TA Instruments mod. Q400 (New Castle, DE, USA). The heating program was set from −70 °C to 70 °C, using a constant heating rate of 2 °C min^−1^. Rectangular samples with dimensions 10 × 10 × 4 mm^3^ were subjected to a constant force of 0.02 N.

### 2.7. Water Uptake of PHBH-ASF/OLA Composites

The water absorption study was carried out according to the method described in ISO 62:2008, with distilled water at 23 ± 1 °C, for 9 weeks. The specimens had rectangular dimensions of 80 × 10 × 4 mm^3^. Before starting the immersion, samples were dried at 60 °C for 24 h in an air circulating oven, model 2001245 DIGIHEAT-TFT from J.P. Selecta, S.A. (Barcelona, Spain).

Samples were extracted periodically from the water every planned period. They were dried to remove any remaining surface moisture and weighed on a precision analytical balance model AG245, from Mettler-Toledo Inc. (Schwerzenbach, Switzerland). After this measurement, they were re-immersed in the distilled water bath.

The amount of absorbed water during the water uptake process can be calculated following this expression:(2)$$\Delta m_{t}\left(\%\right)=\left(\frac{W_{t}-W_{0}}{W_{0}}\right)\times 100$$
where *w_t_* stands for the sample weight after an immersion time of t; *w_0_* corresponds to the initial weight of the dried, before the immersion.

ISO 62:2008 establishes the application of first Fick’s Law to determine the diffusion coefficient, *D*, from the collected data regarding the increase of mass by immersion, by means of the expression (2). The calculation of *D* can be done in the linear zone of the water absorption plot. In this initial stage, *w_t_*/*w_s_* is a linear function Δ*m_t_* = f($\sqrt{t}$), that allows to determine *D* from the slope, *θ* [21,22,23,24].
(3)$$\frac{W_{t}}{W_{S}}=\frac{4}{d}{\left(\frac{D\;t}{\pi }\right)}^{\frac{1}{2}}$$
where *D* represents the coefficient of diffusion, *d* stands for the initial thickness of the specimen and *w_s_* stands for the saturation mass in the linear zone. If we plot *w_t_*/*w_s_* against $\sqrt{t}$, it is possible to obtain the slope (*θ*) as this condition is met, *w_t_*/*w_s_* (≤0.5), then the *D* value can be calculated by following the above-mentioned expression [25].

A correction (Stefan’s approximation) is applied to this calculation for the exact calculation of the *D* according to the dimensions of the specimens:(4)$$D_{c}=D{\left(1+\frac{d}{h}+\frac{d}{w}\right)}^{-2}$$
where *D_c_* is the geometrically corrected diffusion coefficient, *h* is the length, *w* is the width of the sample and *d* is the thickness. This equation is based on the assumption that the diffusion velocities are the same in all directions [23,24,25].

## 3. Results and Discussion

### 3.1. Mechanical Properties of PHBH-ASF/OLA Composites

Table 2 shows the results obtained from mechanical characterization (tensile test, hardness and impact Charpy) of PHBH-ASF/OLA composites. The addition of ASF to the PHBH matrix resulted in composites with greater stiffness with increasing ASF wt %. With only 10 wt % ASF, the elastic modulus in tensile test (*E_t_*) increases to 1310 MPa from 1065 MPa (neat PHBH without lignocellulosic filler). This means an increase of 23%. This % increase is, obviously higher, for PHBH composites containing 30 wt % ASF. Regarding the maximum tensile strength (*σ_max_*), the incorporation of natural fillers to the polymeric PHBH matrix, promotes a noticeable decrease. Neat PHBH offers a tensile strength of 20 MPa, which decreases to 16 MPa with only 10 wt % ASF and to 12 MPa with 30 wt % ASF. Singh et al. [4] established that the decrease of tensile strength results from stress concentration at the polymer/filler interfaces. There is a lack of interface interactions between the polymeric matrix (highly hydrophobic) and the lignocellulosic particles (highly hydrophilic), which gets more pronounced with increasing particle content [4]. The mechanical behaviour of these composites highly depends on the potential interactions between the polymer matrix and the surrounding lignocellulosic filler/particle. The lack of (or poor) adhesion leads to formation of microscopic gaps that are responsible for a discontinuous material with the subsequent stress concentration phenomenon [26].

Furthermore, as usual in composites with lignocellulosic fillers/reinforcements [27,28,29,30,31,32,33,34], the plastic deformation capacity of WPCs decreases in a dramatic way. If we focus on the elongation at break (*ε_b_*), the intrinsic very low *ε_b_* values for neat PHBH (around 8% after an aging time of 15 days), are reduced to half with 20 wt % ASF. Composites with 30 wt % ASF, has a *ε_b_* of only 3.5%. This means a much more fragile and less resistant behaviour of PHBH-ASF composites with increasing wt % ASF without any other component. This mechanical behaviour is like those obtained in other thermoplastic matrix composite systems with natural fillers [3,26,35,36,37]. Nevertheless, the addition of OLA leads to an improvement of the elongation at break due to a compatibilization between PHBH/ASF by the interaction of compatibilizer with terminal groups of PHBH and lignocellulosic particles. Additionally, a plasticization effect on the matrix can be expected by OLA acting as lubricant inside de polymer chain. Both effects were reported by Quiles-Carillo et al. with different biobased and petroleum-derived compatibilizers on PLA/ASF [38].

With respect to the hardness values, the increase in the wt % ASF content favours a slight increase in Shore-D hardness as expected since tensile characterization suggested increased stiffness. In fact, the Shore-D hardness increases from 60.2 (neat PHBH) to 66.2 (composite containing 30 wt % ASF) [30]. On the other hand, the impact resistance is one of the properties with the greatest decrease in uncompatibilized PHBH-ASF composites. First, it should be noted that PHBH is a thermoplastic with an intrinsically low impact resistance. This fragile behaviour of PHBH was greatly affected by the addition of ASF (even with low wt % ASF content. The results show how the addition of 10 wt % ASF, decreases the impact strength to values lower to the half (1.8 kJ m^−2^) of neat PHBH (4.3 kJ m^−2^).

It is worthy to note that WPCs are widely used in applications that include fencing, garden objects, furniture, decking, and so on. The technical requirements will depend on the final application. Considering the mechanical results, these WPCs offer relatively low tensile strength and low elongation at break even without ASF filler. The addition of ASF up to 30 wt % and OLA as compatibilizer, gives interesting materials with a wood-like appearance but they cannot be used for medium technological applications as mechanical properties are low. Moreover, addition of 30 wt % ASF leads to a cost-effective material as PHBH matrix is still an expensive bacterial polyester.

The impact resistance values dropped down to similar values with increasing wt % ASF content. For 30 wt % ASF the absorbed-energy per unit area is around 1.6 kJ m^−2^, which represents a loss of almost 63% of the capacity to absorb energy during impact conditions, which is representative for the overall toughness. These results showed a clear embrittlement and loss of toughness on PHBH-ASF composites as the wt % ASF content increases. The small lignocellulosic ASF particles form a dispersed phase in the thermoplastic matrix (due to the high hydrophilicity of ASF particles, it is possible to form aggregates which lead to worse properties). This dispersed phase interrupts the continuity of the PHBH matrix; in these conditions, the stress transfer between the particle and the matrix is not allowed. In addition, as observed in Figure 1, ASF particles are not spherical; their shape is very irregular (with angular shapes) and could act as micro-notches that promote formation of microcracks and subsequently affect the crack growth. This phenomenon justifies the decrease in toughness in PHBH-ASF composites [3,4,29,37,39]. Furthermore, since the polymeric matrix is non-polar (hydrophobic) and the ASF particles are highly polar (highly hydrophilic due to its lignocellulosic composition), there is no (or very poor) matrix–particle interaction along the interface. This lack of interface causes a fragilizing effect by concentrating the stresses and decreasing the potential plastic deformation capacity in PHBH-ASF composites [26,35,40,41,42,43].

Figure 2 shows in a detailed way the lack of interface between PHBH and ASF particles. Figure 2a,b show a clear micro-gap surrounding the ASF particle. This gap sizes range from 1 μm to 3–4 μm (see white arrows), and the gaps are responsible for lack of interactions in the polymer-particle interface. This gaps are responsible for interrupting the continuity in uncompatibilized PHBH-ASF composites and do not allow stress transfer. This suggests ASF particles do not act as reinforcing material; furthermore, they promote stress concentration leading to a brittle material. Despite this, addition of an oligomer of lactic acid (OLA), could potentially provide improved toughness as observed in Figure 2c,d, which corresponds to the compatibilized composite with 10 *phr* OLA. At higher magnifications ASF particles are completely embedded by the PHBH-rich matrix (Figure 2d, see white arrow with a circle end). This situation is similar to that obtained in composites with 20 *phr* OLA (Figure 2e,f) which shows absence of gap between the ASF particles and the surrounding matrix. This oligomer has carboxylic acid and hydroxyl terminal groups that can readily react (interact) with hydroxyl terminal groups in PHBH and, obviously, with the hydroxyl groups in ASF (mainly, cellulose, lignin and hemicelluloses). As can be seen, the gap is remarkably reduced (white arrow) and this could contribute to improve toughness and stress transfer [44,45]. Despite high polarity ester groups can establish somewhat interactions with polar groups in cellulose, the main compatibilizing effects are obtained with high reactive groups such as maleic anhydride, carboxyl acids, end-chain hydroxyl groups, glycidyl methacrylate as reported by Pracela et al. [46] by using functionalized copolymers to provide increased interface interactions between a polymer matrix and cellulose particles. Some interactions between ester groups and cellulose particles have been described by Chabros et al. [47] in thermosetting unsaturated polyester resins with cellulose fillers; in particular they describe some interactions between the polar ester groups and hydroxyl groups in cellulose by hydrogen bonding. These small range interactions can also occur in PHBH/ASF composites, but their intensity is lower than that provided by the reaction of carboxylic acid and hydroxyl terminal groups in OLA with both hydroxyl groups in cellulose and PHBH through condensation or esterification reactions. As described by Mokhena et al. [48] the ester groups in PLA are not enough to provide intense interactions between the polyester-type matrix and the cellulose filler. They report the need of different treatments on cellulose such as acetylation, glyoxalization, silylation, treatment with glycidyl methacrylate (GMA), among others to improve polymer-matrix interactions. This could be related not only to the polarity but also with the hydrophilic nature of ASF and the hydrophobic nature of PHBH.

The PHBH-ASF composite with 30 wt % ASF, showed the worst mechanical properties in terms of ductility and toughness. This was taken as a reference material to improve its properties by the addition of a compatibilizer/plasticizer. An oligomer of lactic acid ester (OLA) was added in different proportions (10 and 20 *phr*) to provide compatibilization and some plasticization [49,50]. In Table 2, the increase in impact resistance for the reference uncompatibilized composite is observed with the addition of 10 *phr* and 20 *phr* of OLA. The addition of small amounts (10 *phr* ≈ 0.1 wt %) of this oligomer significantly improves the impact resistance of the composite [51]. It changes from 1.6 kJ m^−2^ (uncompatibilized PHBH-ASF with 30 wt % ASF) to 2.4 kJ m^−2^ for the same composite with 20 *phr* OLA, which represents a % increase of 33%. Considering that this increase in toughness is related to an improvement in ductility, subsequently, Shore-D hardness values decreased, changing from 66.2 Shore-D to 58.6 and 50.0 Shore-D for 10 *phr* and 20 *phr* OLA content, respectively.

This improvement on toughness is corroborated by the capacity of deformation observed in compatibilized composites. By adding only 10 *phr* OLA to the reference uncompatibilized PHBH-ASF composite, its elongation at break is almost doubled. The compatibilization effect reported in FESEM images (gap reduction) provided a more efficient load transfer between PHBH and ASF leading to an improvement of elongation at break as Quiles-Carrillo et al. reported [38]. With 20 *phr* OLA, the *ε_b_*, increase up to 9.7%, which means an increase of 177% compared to the same composite without OLA. However, the most striking thing is that the *ε_b_* with 20 *phr* OLA is even higher than that of neat PHBH which is a very positive feature, mainly in this highly brittle material. The composite with 30 wt % ASF and 20 *phr* OLA shows a *ε_b_* value of almost 20% higher than neat PHBH without any filler. These results indicate a marked plasticizing effect of this OLA oligomer, which is corroborated by the values of the tensile strength (*σ_t_*) and the elastic modulus (*E_t_*). The incorporation of short-chain oligomers OLA increases the free volume of the polymer chains in PHBH, which leads to a reduction in the stiffness and an increase in the ductility of composites. The improvement in ductility these PHBH-ASF composites with the addition of OLA, produces a decrease in *σ_t_* to 8 MPa with 20 *phr* OLA, which is slightly lower than the *σ_t_* compared to the same composite without OLA (12 MPa). On the other hand, the decrease in the *E_t_* observed by OLA addition, indicated that the compatibilized composites are not as rigid as uncompatibilized materials. The obtained *E_t_* values for 10 and 20 *phr* OLA are 1158 MPa and 735 MPa respectively. This represents a decrease of 33% and 58%, with regard to the reference uncompatibilized composite with an *E_t_* value of 1744 MPa [51].

### 3.2. Thermal Properties of PHBH-ASF/OLA Composites

A comparative plot of the DSC thermograms is represented in Figure 3 and the main thermal parameters are summarized in Table 3. All thermograms are characterized by a first change at very low temperature (around 0 °C) in the corresponding baseline, which is attributable to the corresponding glass transition temperature (*T_g_*). Neat PHBH is a thermoplastic with low *T_g_*, close to 0 °C; similar values to this have been reported in several studies with PHBH [14]. In a first analysis, it was determined that the addition of lignocellulosic filler, ASF, to PHBH, slightly decreases the *T_g_* to values comprised in the −0.5–1.9 °C range for all uncompatibilized composites. Considering that *T_g_* are not a unique temperature, but a temperature range in which the material undergoes a change from a glassy state to a rubbery state, it can be assessed that these slight variations in *T_g_* are not significative. The marked exothermic peak observed in the DSC thermogram of neat PHBH corresponds to the cold crystallization phenomenon, and its peak (corresponding to the temperature in which the crystallization rate is maximum) is located at 54.6 °C (*T_cc_*). The addition of lignocellulosic ASF particles does not influence the *T_cc_* of PHBH in the developed composites. It can only be observed a dilution effect (the cold crystallization peak height is smaller in PHBH/ASF composites, compared to neat PHBH; this is because the PHBH/ASF composite contains 30 wt % ASF which has no thermal transition in this temperature range, and consequently, the intensity of the peak is lower). At higher temperatures, the DSC thermograms show three small and broad endothermic peaks corresponding to the melting process of PHBH [52]. As already indicated in other studies [16,18,19,53,54,55], due to the polymorphism of the PHBH crystals during the crystallization process, its melting occurs at different temperatures. This situation of PHBH, is identical to other polyhydroxyalkanoates (PHAs) which present three melting peaks at different temperatures *T_m1_, T_m2_* and *T_m3_*, too. Neat PHBH used in this work, show three melting peak temperatures located at 111 °C, 130 °C and 162 °C, respectively. The addition of lignocellulosic ASF particles does not produce significant changes in melting temperatures as observed in other studies [36]. The analysis of the enthalpies corresponding to the cold crystallization process (Δ*H_cc_*), indicated that the highest enthalpy corresponded to neat PHBH during the second heating cycle (Δ*H_cc2_* = 26.7 J g^−1^). These values decreased gradually with the addition of ASF particles, down to values of 3.7 J g^−1^ for the sample with 30 wt % ASF. The dilution effect (which means considering the actual PHBH content without taking into account the wt % ASF for the cold crystallization enthalpy calculation), would give a theoretical diluted enthalpy of 18.69 J g^−1^, which is remarkably higher than the actual obtained value of 3.7 J g^−1^. These differences are not so pronounced for 10 and 20 wt % ASF.

In a second stage, an oligomer of lactic acid (OLA) was added to improve toughness of the PHBH-AS composite with worst toughness, i.e., composite with 30 wt % ASF. The DSC thermograms, (Figure 3), show a similar thermal behaviour to the OLA-free composites. The addition of 10 *phr* and 20 *phr* OLA to this composite, shows a decrease of the PHBH *T_g_* down to values around −5 °C, with respect to −1 °C for the same composite without OLA. The low molecular weight OLA chains offer slightly increased mobility in comparison with the polymer chains of neat PHBH; similar results were reported by Quiles-Carrillo et al. [56]. This phenomenon increases the free volume in the polymeric structure and leads to a poor plasticizing effect. It is true that addition of oligomers of lactic acid to PLA, usually leads to a remarkable decrease in PLA’s *T_g_* as reported by Burgos et al. [57] from 66 °C (neat PLA) down to −10 °C. Nevertheless, the decrease in *T_g_*, is directly related to the PLA and OLA structure. D. Lascano et al. [51] reported a decrease of PLA *T_g_* from 63.3 °C (neat PLA; different grade) down to 50.8 °C with 20% OLA (different commercial grade of OLA). Armentano et al. [58] reported a dual plasticization effect of PHB and OLA on PLA, but the decrease in *T_g_* was not as important as the above-mentioned by Burgos et al. Moreover, Amor et al. [59], reported a slight plasticization effect on PLA/PHBH blends by using OLA in pellet form with a dual plasticization effect of PHBH (which provided a 3 °C decrease in *T_g_* of PLA with a PHBH loading of 10 wt %) and OLA (which provided a decrease of 1 °C with a load of 1 wt %). All these results show the disparity in plasticization of PLA with OLA even they share the same chemical structure. Therefore, the plasticization effect on PHAs is even more complex and has not been studied previously independently of blends with PLA. In this work, this decrease in *T_g_* values are very low, but we must bear in mind that PHBH structure contains medium chain hydroxyalkanoates and these, contribute to lowering crystallinity compared to P3HB. Therefore, this slight decrease could be representative of somewhat plasticization effect provided by OLA. Besides this, it corroborates the mechanical results analyzed previously [31,40,51]. *T_cc_* values increased from 52 °C for the sample without OLA, to 62 °C for the sample with 20 *phr* OLA. This is not the typical effect of a plasticizer which increases chain mobility and, therefore, the cold crystallization process is shifted to lower temperatures as reported by Lascano et al. [51] and Ferri et al. [60]. Nevertheless, some additives such as maleinized linseed oil (MLO), maleinized cottonseed oil (MCSO), or even epoxidized vegetable oils, promote an overlapping of several phenomena such as slight plasticization, chain extension, branching and, in some cases, potential crosslinking due to the multifunctionality of these compounds. Some of these vegetable-oil derivatives, produce the same behaviours as observed in this study, i.e., a shift of the cold crystallization process to higher temperatures, due to the disrupted overall structure they provide with branches, chain extension, and so on, as reported by Garcia-Campo et al. [61]. Limiñana et al. [62] reported the potential of these modified vegetable oils as compatibilizers for PBS and lignocellulosic fillers, due to reaction of oxirane, maleic anhydride groups with hydroxyl groups in almond shell flour. In this study, it seems OLA has a similar effect to those modified vegetable oils, since changes in *T_g_* are very low (which is representative for very slight plasticization effect, almost inexistent, since the technique itself has some uncertainty in the obtained values depending on the sample size, geometry, surface contact, and so on), and the cold crystallization is shifted to higher temperatures, thus indicating that other phenomena could be occurring, such as compatibilization and/or chain extension.

The Δ*Hcc_2_* values of PHBH in OLA-compatibilized composites is 11.4 and 15.9.6 J g^−1^, for 10 *phr* and 20 *phr* OLA respectively. It is a striking fact that the same composite without OLA shows a much lower Δ*H_cc2_*, 3.7 J g^−1^. With respect to *T_m1,_ T_m2_* and *T_m3_* it was observed that these thermal transitions were slightly decreased with the addition OLA, which could be related to more or less perfect crystals [51].

On the first heating scan the polymer has been recrystallized at 25 °C for 15 days. Consequently, during the first heating scan the cold crystallization peak did not appear which means that the polymer structure was not able to form crystallites. Under this conditions PHBH reaches a *X_c1_* value of 13.9%. This increases with the amount of ASF filler up to 15.5% with 20 wt % ASF which suggests that ASF (mainly crystalline cellulose fractions) acts as a nucleant agent [62]. Furthermore, with the addition of 30 wt % ASF, *X_c_*_1_ decreases to 8.3% due to the decrease of free volume necessary for nucleation of polymer as Thomas et al. reported [63]. Mechanical characterization shows no correlation between the degree of crystallinity, while elastic modulus increases up to 30 wt % ASF, the degree of crystallinity is saturated with only 20 wt %. The second scan was performed after a controlled cooling process of 10 °C min^−1^, as a result in the second scan the polymer was able to form crystallites due to a cold crystallization process. Under this condition the degree of crystallinity could increase until 30 wt % ASF. The compatibilizing effect of OLA in both conditions decreased the degree of crystallinity by reducing the gaps between the filler and the matrix as it is reported in FESEM analysis and Gong et al. proposed [64].

Thermogravimetric analysis, TGA, allowed to analyze the thermal stability of PHBH-ASF composites. The TGA curves of the studied materials are gathered in Figure 4, and the main thermal degradation parameters are summarized in Table 4. The thermal degradation process of neat PHBH occurs in a single step. PHBH shows good thermal stability up to 266.8 °C (the onset degradation temperature was taken as the temperature for a weight loss of 2%, and it is denoted as *T_2%_*) of PHBH. Above this temperature, thermal degradation starts, with a very fast weight loss and a temperature of maximum degradation rate, *T_max_*, of 308.9 °C, obtained from peak corresponding to the first derivative of its TGA curve or first DTG (Derivative thermogravimetry) (Figure 4b). The results obtained by Singh for PHBV indicated that the degradation process involves the breaking of polymer chains and hydrolysis. Since PHBH presents a similar structure, the mechanism of degradation should be similar [4]. Reaching the endset of the degradation process, located at 371 °C, PHBH generates a small residue or ash of 2.4 wt % of its initial weight. These results are in accordance to those obtained in other works [14,65].

The thermogram obtained for ASF particles shows different degradation processes corresponding to three different sections [31,35,37]. Since it is an agro-food waste of lignocellulosic nature, it shows a first weight loss around 100 °C, which corresponds to the loss of remaining water in ASF, specifically 6.3 wt %. During the dynamic degradation process, when temperature reaches 213 °C, a rapid weight loss is observed in two main steps. The first step of weight loss, of about 44.2 wt %, corresponds to the degradation of the cellulose and hemicellulose contained in ASF particles. The first component to start degradation is hemicellulose, followed by cellulose and lignin. Lignin shows a slower (in a wide temperature range) degradation process, so the third section of the TGA curve shows a lower slope, starting at 357.3 °C (temperature change of slope in the thermogram) up to 500 °C, with a loss of 47.6 wt %. Complete degradation occurs around 500 °C, leaving a final carbonaceous residue or ash of 1.5 wt %, mainly from lignin. In Figure 4b, it can be seen how the temperature corresponding to the maximum degradation rate of hemicellulose-cellulose fraction is located at 300.6 °C, while the lignin fraction maximum degradation rate is close to 460.7 °C. Perinovic determined that degradation of polysaccharides, hemicellulose and cellulose starts between 220–290 °C, while lignin degradation range is comprised between 200 °C and 500 °C [32,33,35,37,41,66].

The TGA curves of PHBH-ASF composites indicated that the thermal degradation process is a linear combination of the two individual degradation phenomena observed in PHBH and ASF, the first part of the curve is identical to neat PHBH, and a small hump at the end of the curve can be detected, which is attributable to residual degradation of ASF which changes with the ASF wt % [4,31]. For any wt % content in almond shells, the TGAs are characterized by presenting degradation start temperatures (*T_2%_*) slightly lower than neat PHBH (with a *T_2%_* of 286.8 °C), in the 220–250 °C temperature range. The addition of lignocellulosic fillers leads to slightly lower thermal stability, since ASF particles degrade separately from PHBH, and the overall effect is the PHBH-ASF composites has reduced its thermal stability. This factor is due to the initial degradation of low molecular weight components on almond shell flour such as hemicelluloses. Quiles-Carrillo et al. [38] reported similar results with PLA composites with almond shell flour.

TGA curves of composites are very similar to those of the PHBH (as it is the main component), with practically only one step degradation stage, but with a small hump at higher temperatures, corresponding to lignin degradation. As the ASF content increases, this hump becomes more pronounced. This process ends at temperatures around 500 °C, generating small amounts of residue close to 2 wt %. The TGA curves of PHBH-ASF composites indicated that from a thermal point of view, the incorporation of lignocellulosic fillers such as ASF, slightly reduces the stability, but even in this case, the processing window is not compromised since all the onset degradation temperatures are above 250 °C, and the recommended processing temperature for this polymer is 140–150 °C.

On the other hand, composites with 10 *phr* and 20 *phr* OLA, show a very similar thermal degradation behaviour to the reference uncompatibilized composite (PHBH-ASF with 30 wt % ASF) without OLA. The addition of 10 *phr* OLA seems to slightly improve the thermal stability of the developed composites. The characteristic thermal degradation values are delayed by 34 °C (onset degradation temperature, *T_2%)_* and by 10 °C (for the maximum degradation rate, *T_deg_*) compared to the unmodified reference composite without OLA. The biggest improvement in thermal stability is observed for the composite with 10 wt % OLA. This improvement is due to the chemical interaction of the compatibilizer with both components of the composite as above-mentioned. The complex structure formed after reaction of OLA with both PHBH and ASF, can act as a physical barrier that obstructs the removal of volatile products produced during decomposition [67].

### 3.3. Thermomechanical Properties of PHBH-ASF/OLA Composites

Figure 5a shows the variation of the flexural single cantilever) storage modulus, *E’*, with respect to temperature, obtained by DMTA analysis. It can be seen graphically how *E’* decreases with increasing temperature in all the developed composites, as expected due to the softening of the polymeric PHBH matrix. At low temperatures, *E’* values are high in all composites, since this temperature range corresponds to the elastic-glassy behaviour of the PHBH matrix. In this first zone, *E’* for neat PHBH is 1869 MPa at −40 °C, which is lower than *E’* values of any of the uncompatibilized PHBH-ASF composites (e.g., *E’* is 2019 MPa for the PHBH-ASF composite with 30 wt % ASF at −40 °C). These results are in accordance with those obtained by mechanical characterization which suggested a stiffening as the wt % ASF increased [37]. Table 5 shows the numerical comparison of the variation of *E’* as a function of ASF wt % and OLA *phr*, at two different temperatures.

As the temperature increases, *E’* decreases rapidly as it acquires a rubbery state behaviour. This is related to the α-relaxation process or the glass transition temperature (*T_g_*). Table 5 shows how at 25 °C, the *E’* value for neat PHBH has decreased to 1345 MPa from 1869 MPa at −40 °C and, subsequently, the stiff-elastic behaviour changes to a rubber-like behaviour. The same trend can be observed for uncompatibilized PHBH-ASF composites. However, when compared to the neat PHBH matrix, at the same temperature, the higher the wt % of ASF, the stiffer the composite becomes. With only 10 wt % ASF, *E’* at 25 °C increases 6.4% with respect to PHBH at the same temperature. Accordingly, the composite with 30 wt % ASF offers higher stiffness (a percentage increase of 19% regarding neat PHBH). The presence of ASF particles finely dispersed in the PHBH matrix restricts the mobility of the polymer chains, thus decreasing their viscous behaviour, which causes an increase in the *E’* value as the ASF loading increases [4,27,28,29,39]. In addition, at this temperature range, lignocellulosic components are below its *T_g_*, which means they show a glassy behaviour that promotes increased stiffness. Nevertheless, it is important to bear in mind that the main component in PHBH/ASF composites is PHBH and the dynamic behaviour is highly influenced by PHBH behaviour since conditions are not as aggressive as a conventional tensile test up to fracture. These results corroborate those obtained in the mechanical characterization of PHBH-ASF composites.

As in previous analyses, by adding small amounts of OLA to the PHBH-ASF composite with the highest ASF loading, which shows the worst ductile/toughness properties, the DMTA graphs in Figure 5a show the lowest *E’* in the temperature range analyzed. At −40 °C, *E’* decreases from 2019 MPa without OLA addition, to 1601 MPa and 853 MPa for the addition of 10 *phr* and 20 *phr* OLA, respectively. The trend is the same at room temperature (25 °C). With an addition of only 10 *phr* OLA the *E’* is decreased by 16%, and with 20 *phr* OLA *E’* is reduced even more, thus decreasing the rigidity of PHBH-ASF composites and, subsequently their *E’* [40]. It is worth noting the extremely small changes in *T_g_* obtained by DMTA which suggests, as DSC, very poor plasticization effect, suggesting compatibilization is one of the most representative effects of OLA in this PHBH/ASF system.

Figure 5b shows the variation of the dynamic damping factor (*tan δ*) as a function of temperature, for neat PHBH, uncompatibilized PHBH-ASF and PHBH-ASF/OLA composites. Despite there are several criteria to obtain the *T_g_* from DMTA graphs, the most used is the peak maximum of *tan δ*. The *T_g_* values obtained for all developed materials are gathered in Table 5. It can be observed that *tan δ* peaks are slightly moved towards higher temperatures in uncompatibilized PHBH-ASF composites, compared to neat PHBH (a maximum shift of 3–4 °C). This change is not significant as observed in other techniques such as DSC. On the other hand, the addition of OLA leads to slightly lower *T_g_* values by DMTA, but once again, these changes are not high enough to give a clear evidence of the chain mobility restriction by ASF particles or increased chain mobility by OLA.

For potential structural/engineering/conventional applications of WPCs, it is very important to know their dimensional stability with temperature. This can be assessed by thermomechanical analysis (TMA) which allows us to obtain the coefficient of linear thermal expansion (*CLTE*). It must be stated that a good dimensional stability involves low *CLTE* values. Table 6 summarizes the CLTE values for the developed composites, at temperatures below and above their corresponding *T_g_*. In general, at temperatures below *T_g_* the *CLTE* values are much lower than above their *T_g_*. The dimensional expansion of the material is lower at low temperatures because the material is more rigid, which is the typical glassy behaviour below *T_g_*. Above *T_g_*, the behaviour is viscous or rubber-like and so that, the dimensional expansion is favoured, with higher *CLTE* values.

First, the analysis of the *CLTE* values at temperatures below *T_g_* shows that increasing wt % ASF gives more dimensional stability, which is in accordance with previous mechanical results that suggested a clear stiffening with ASF addition. Pure PHBH has an initial value of 77.1 μm m^−1^ °C^−1^, which decreases to 66.8 μm m^−1^ °C^−1^ with 30 wt % ASF particles (which represents a % decrease of 13%), which involves improved dimensional stability. However, OLA addition significantly increases the *CLTE* values, as typical plasticizers do, thus leading to slightly lower dimensional stability. With 10 *phr* and 20 *phr* OLA, *CLTE* is 72.0 and 90.7 μm m^−1^ °C^−1^, respectively.

Secondly, the results obtained in the study at temperatures above *T_g_*, show the same tendency as the results discussed in the previous paragraph. Neat PHBH shows an initial *CLTE* of 160.7 μm m^−1^ °C^−1^ (much higher than below *T_g_*), which decreases to 140.3 μm m^−1^ °C^−1^ with 30 wt % ASF particles. Identically as observed previously, *CLTE* becomes greater again with the addition of OLA, reaching values of 194.3 μm m^−1^ °C^−1^ with 20 *phr* OLA. In general, as the ASF content increases, composites show improved dimensional stability [4,29,68]. However, these dimensional expansions are higher than those of neat PHBH in PHBH-ASF composites (30 wt % ASF) with 10 *phr* and 20 *phr* OLA, due to the plasticizing effect [51].

### 3.4. Evolution of the Water Uptake and Water Diffusion Process in PHBH-ASF/OLA Composites

The water absorption capacity of WPCs is an important feature in some applications due to the lignocellulosic component. This creates a three-dimensional path inside the polymer matrix that allows water entering (for example when the composite is subjected to high relative humidity environments) and this causes an expansion. It is possible that after this initial stage, this WPC could be subjected to drying at sun with low humidity; then this 3D-path allows water/moisture removal, promoting a contraction. This situation is quite usual in WPCs such as those used in fences, decking, and so on. This repeated expansion–contraction cycles could lead to formation of microcracks. Figure 6 shows the mass increase (wt %) with respect to immersion time in water for the PHBH-ASF/OLA composites. It can be seen graphically that during the first week of immersion, the developed composites show a rapid increase in mass by water absorption (Δ*mass*). As immersion time increases, the mass increase is slower. Some samples even show an asymptotic behaviour, which indicates that saturation has been reached (Δ*mass∞*). This type of behaviour corresponds to that indicated by the first Fick’s Law.

Obviously, due to the hydrophobicity of neat PHBH, it shows the lowest water absorption for nine weeks. After 35 days of immersion it reaches a constant saturation mass, Δ*mass∞* of 0.53%, which is maintained practically until 63 days of immersion. As mentioned above, the *X_c_* of neat PHBH is 13.9%. The addition of ASF (highly hydrophilic particles due to its composition: cellulose, hemicellulose and lignin as the main components) considerably increases water absorption, but the effects of PHBH crystallinity can also be observed in this behaviour. The composite with 10 wt % ASF reaches a water saturation mass, Δ*mass∞* of 1.46 wt % after 42 days (*X_c_* of PHBH in this composite is 14.9% and this prevents from water entering). In a similar way, uncompatibilized composites containing 20 wt % show a relatively low Δ*mass∞* of 3.1 wt %. This is almost double the previous value (with 10 wt % ASF). This value is expected since the *X_c_* of PHBH in this composite is close to 15.5%. Nevertheless, the composite containing 30 wt % ASF, shows a remarkable increase in Δ*mass∞* up to values of 7.1 wt % after nine weeks, which represents almost 14 times the value for neat PHBH, representing a typical result in most WPCs [23,36,43]. The result is higher than the extrapolation from the ASF wt % (which should be of about three times the value of the composite with 10 wt % ASF, i.e., 4.5 wt %); despite this, we have to bear in mind that the *X_c_* of PHBH in this composite has decreased to 8.3% and this has a negative effect on water absorption as amorphous regions allow water entering [69,70]. Therefore, the increase in water absorption is not only related to the number of lignocellulosic components, but also with the degree of crystallinity of the polymer matrix. Cellulose promotes water absorption due to hydroxyl (–OH) groups that interact with water molecules [21,22,23,25,41,71].

On the other hand, the addition of OLA to the reference composite (PHBH with 30 wt % ASF), shows an unexpected behaviour. As one can see in Figure 6, the water absorption curve with OLA, moves to lower wt % absorbed water. The values of the mass increase after nine weeks of immersion reach values of 5.8 wt % and 5.5 wt % for OLA contents of 10 *phr* and 20 *phr*, respectively. This means a decrease in water absorption of 18% and 22.6% respectively when OLA is added to the PHBH-ASF system. Typical plasticizers, increase the water absorption as they are responsible for an increase in the free volume, thus allowing water molecules to enter. Nevertheless, OLA not only provides plasticization effects, but also improved polymer-particle interaction among the interface due to the interaction between the hydroxyl groups in OLA and the hydroxyl groups of both PHBH (terminal groups), and cellulose/hemicellulose/lignin in ASF particles. Therefore, in addition to a plasticization phenomenon, we could think on an additional compatibilization effect which was also evidenced in Figure 2 (FESEM characterization), in which the gap size was higher on OLA-free composites than composites with OLA.

Table 7 shows the values of the diffusion coefficient (*D*) or diffusivity of water into the developed composites, by applying the first Fick’s Law. The lowest corrected diffusion coefficient, *D_c_* value is offered by neat PHBH, as expected due to its hydrophobicity. The only addition of 10 wt % ASF, leads to a *D_c_* value, almost triple compared to neat PHBH. Obviously, ASF is responsible for water entering the composite structure; therefore, uncompatibilized composites with 30 wt % ASF, shows an increase of two orders of magnitude. Due to the hydrophilic nature of ASF particles, and possibly accentuated by the capillarity of the micro-gaps between PHBH and the embedded ASF particles, water molecules can easily enter into the composite structure as in most WPCs [51,71]. Finally, the *D_c_* values PHBH-ASF/OLA composites remain with similar values to those of the same composite without OLA. Thus, it was deduced that the amount of wt % ASF is the parameter with the greatest influence on the water diffusion process in PHBH-ASF/OLA composites, together with the degree of crystallinity as previously discussed with the relationship of the water absorption and the wt % ASF loading and the degree of crystallinity of the PHBH matrix. Since the *D_c_* values for the composites with 10 and 20 *phr* OLA are similar to that of the same composite with 30 wt % ASF, it is possible to conclude the poor plasticization effect of this OLA, suggesting, once again, that compatibilization is the main acting mechanisms of OLA on PHBH/ASF composites.

## 4. Conclusions

The results obtained in this study indicate that the analyzed system of poly(3-hydroxybutyrate-co-3-hydroxyhexanoate) (PHBH) and almond shell flour (ASF), is suitable for the manufacture of fully biobased and environmentally friendly Wood Plastic Composites (WPCs). PHBH-ASF composites present a very interesting set of properties for technical applications as wood substitute materials. The characterization of PHBH-ASF composites showed that the addition of lignocellulosic particles of ASF leads to an embrittlement and reduced toughness. These effects are much more evident with increasing the wt % ASF. To overcome or minimize these negative properties, an oligomer of lactic acid, OLA was added to give PHBH-ASF/OLA composites with improved ductile properties and, subsequently, improved toughness. It is worth noting a remarkable increase in impact strength with 20 *phr* OLA. A higher mobility of the PHBH polymer chains by the addition of OLA is the reason for the improvement on toughness, even on composites with 30 wt % ASF.

The study of the PLA-ASF/OLA composites allowed us to obtain good balanced properties and, therefore, these materials can be used in the WPC industry as they are suitable for technical applications that require certain stiffness and thermal stability, in an interesting range of properties depending on the wt % ASF content. Furthermore, the addition of OLA oligomer decreases the water absorption capacity of PHBH-ASF/OLA, thus broadening potential uses in high humidity environments. Finally, its thermoplastic nature allows it to be easily processed by conventional extrusion–injection moulding, and overall, these composites contribute to a sustainable development and a reduction of the carbon footprint as all the used materials are bio-sourced.

## Figures and Tables

**Figure 1 polymers-12-01097-f001:**
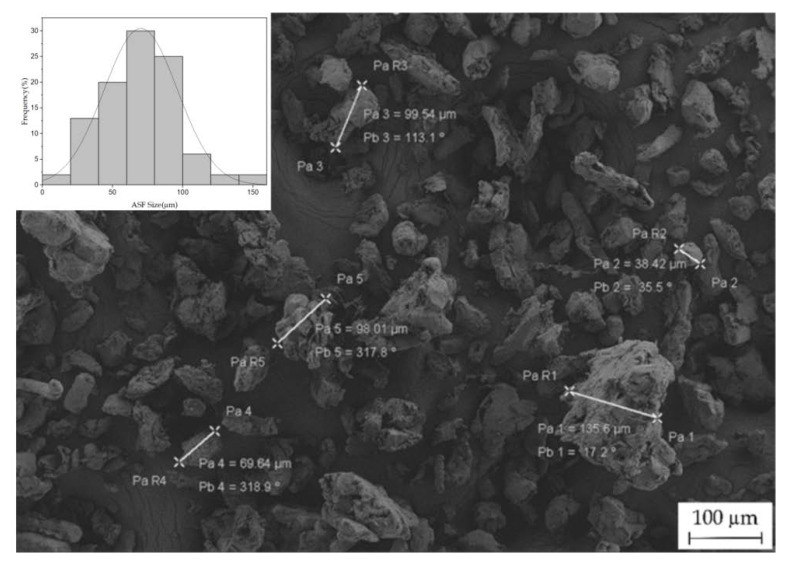
Visual aspect of almond shell flour particles obtained by field emission scanning electron microscopy (FESEM) at 100× and a histogram of their size distribution.

**Figure 2 polymers-12-01097-f002:**
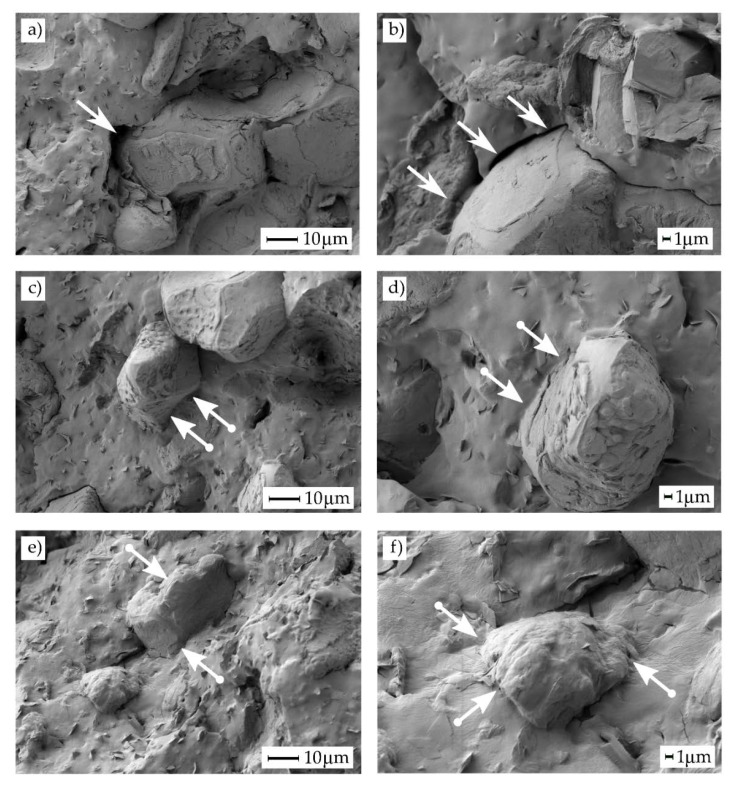
Field emission scanning electron microscopy (FESEM) images at 1000× (left side) and 2500× (right side) corresponding to PHBH-ASF composite with 30 wt % ASF with different OLA content, (**a**) & (**b**) 0 *phr* OLA, (**c**) & (**d**) 10 *phr* OLA, (**e**) & (**f**) 20 *phr* OLA.

**Figure 3 polymers-12-01097-f003:**
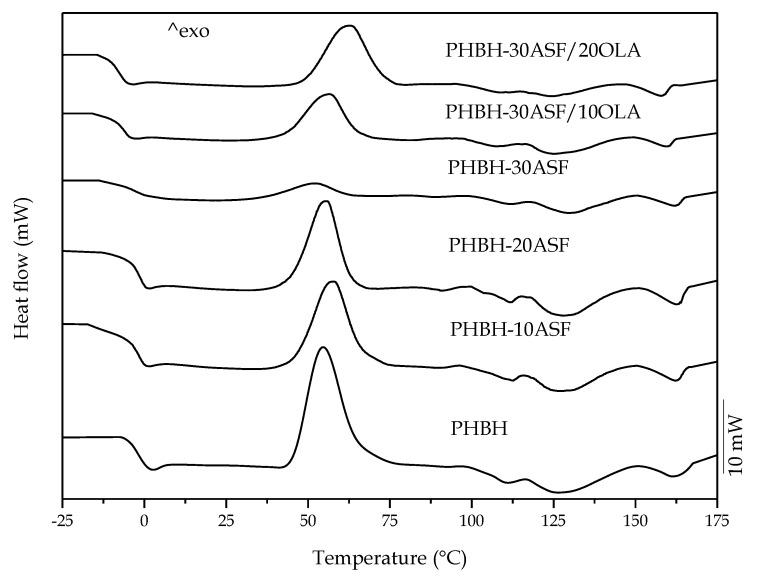
Comparative plot of the second heating curves obtained by dynamic differential scanning calorimetry (DSC) of the different PHBH-ASF/OLA composites with different compositions.

**Figure 4 polymers-12-01097-f004:**
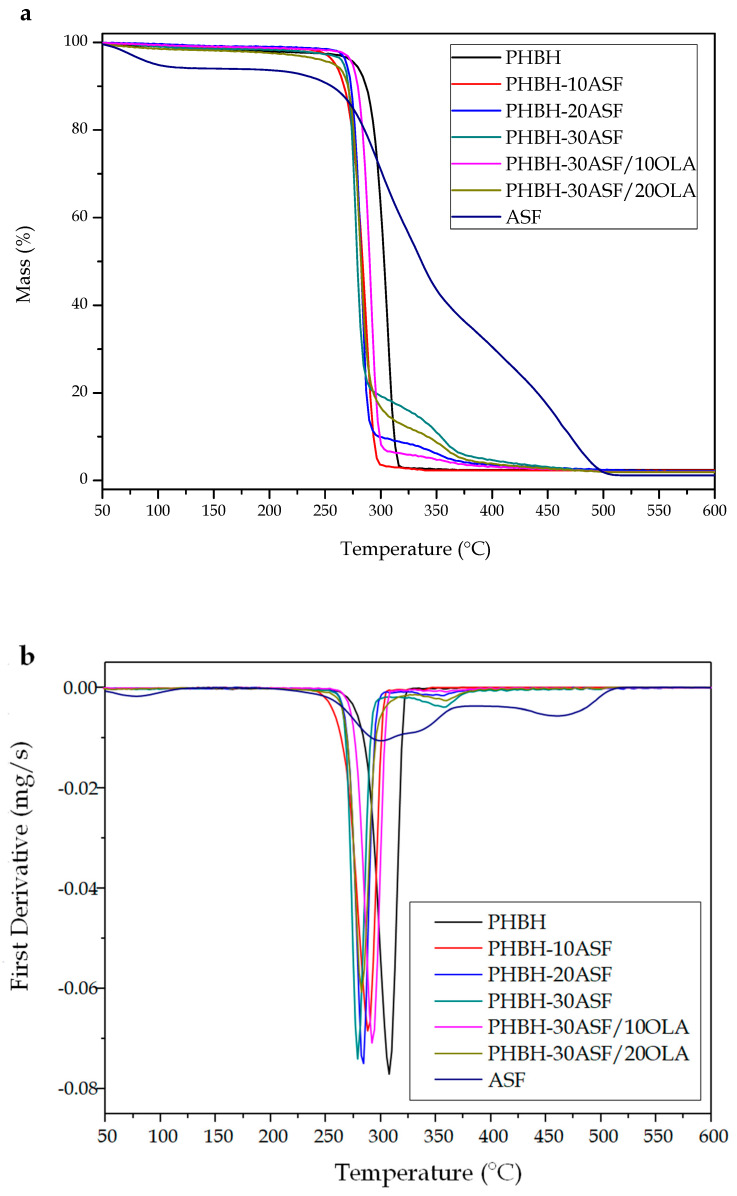
Comparative plot of (**a**) thermogravimetric analysis (TGA) curves and (**b**) first derivative (DTG) of the PHBH-ASF/OLA composites with different compositions.

**Figure 5 polymers-12-01097-f005:**
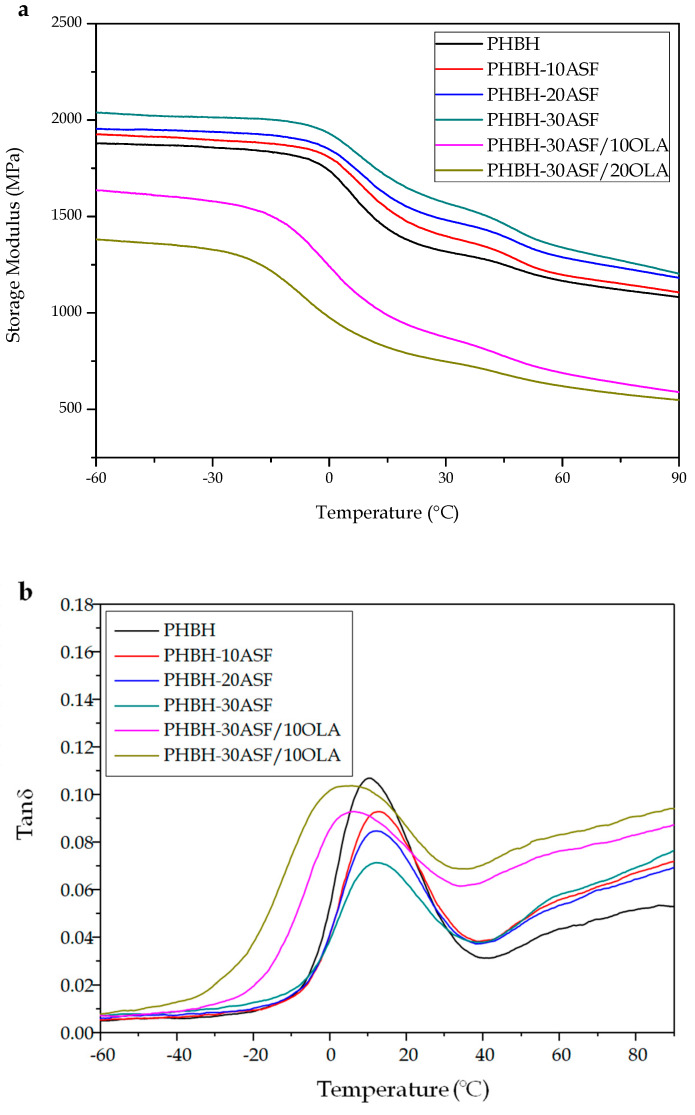
Comparative plot of dynamic-mechanical thermal analysis (DMTA) curves of PHBH-ASF/OLA composites with different compositions: (**a**) flexural storage modulus (*E’*) and (**b**) dynamic damping factor (*tan δ*).

**Figure 6 polymers-12-01097-f006:**
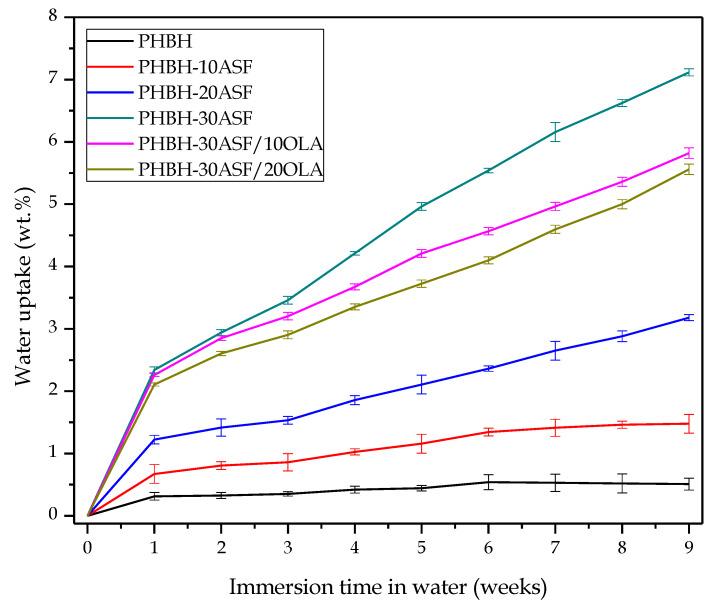
Water uptake of PHBH-ASF/OLA composites with different compositions. Evolution of the water uptake for a period of nine weeks.

**Table 1 polymers-12-01097-t001:** Summary of sample compositions according to the weight content (wt %) of PHBH (Poly(3-hydroxybutyrate-*co*-3-hydroxyhexanoate)), and ASF (Almond Shell Flour) and the addition of OLA (Oligomeric Lactic Acid) as parts per hundred resin (*phr*) of PHBH-ASF composite.

Code	PHBH (wt %)	ASF (wt %)	OLA (*phr*)
PHBH	100	-	-
PHBH-10ASF	90	10	-
PHBH-20ASF	80	20	-
PHBH-30ASF	70	30	-
PHBH-30ASF/10OLA	70	30	10
PHBH-30ASF/20OLA	70	30	20

**Table 2 polymers-12-01097-t002:** Summary of the mechanical properties of the PHBH-ASF/OLA composites with different compositions, in terms of the tensile modulus (*E_t_*), maximum tensile strength (*σ_max_*), elongation at break (***ε****_b_*), Shore-D hardness and impact strength.

Code	Tensile Strength (MPa)	Elastic Modulus (MPa)	Elongation at Break (%)	Hardness Shore-D	Impact Strength (kJ m^−2^)
PHBH	20 ± 1	1065 ± 23	8.1 ± 0.7	60.2 ± 0.2	4.3 ± 0.3
PHBH-10ASF	16 ± 1	1310 ± 35	5.2 ± 0.4	63.5 ± 0.4	1.8 ± 0.2
PHBH-20ASF	14 ± 1	1543 ± 23	4.0 ± 0.4	64.7 ± 0.6	1.7 ± 0.2
PHBH-30ASF	12 ± 1	1744 ± 31	3.5 ± 0.3	66.2 ± 0.6	1.6 ± 0.3
PHBH-30ASF/10OLA	10 ± 1	1158 ± 23	6.2 ± 0.2	58.6 ± 0.5	2.4 ± 0.4
PHBH-30ASF/20OLA	8 ± 1	735 ± 28	9.7 ± 0.8	50.0 ± 0.4	2.9 ± 0.3

**Table 3 polymers-12-01097-t003:** Main thermal parameters of the PHBH-ASF/OLA composites with different compositions, obtained by differential scanning calorimetry (DSC).

Code	*T_g_* (°C)	*T_cc_* (°C)	*T_m1_* (°C)	*T_m2_* (°C)	*T_m3_* (°C)	Δ*H_m1_** (J g^−1^)	Δ*H_cc2_* (J g^−1^)	Δ*H_m2_* (J g^−1^)	*X_c1_* (%)*	*X_c2_ (%)*
**PHBH**	0.3 ± 0.1	54.6 ± 1.1	111.5 ± 1.9	130.8 ± 2.0	162.5 ± 1.2	20.3 ± 0.5	26.7 ± 0.8	33.6 ± 1.3	13.9 ± 1.1	4.7 ± 0.3
**PHBH-10ASF**	−0.5 ± 0.1	57.5 ± 1.8	112.1 ± 1.8	129.7 ± 1.7	162.3 ± 1.8	19.6 ± 0.4	17.4 ± 0.4	29.0 ± 2.2	14.9 ± 1.1	8.8 ± 0.4
**PHBH-20ASF**	−1.9 ± 0.2	55.5 ± 1.9	111.5 ± 2.0	129.6 ± 2.1	163.4 ± 1.4	18.1 ± 0.3	15.0 ± 0.1	26.4 ± 2.1	15.5 ± 0.8	9.8 ± 0.7
**PHBH-30ASF**	−1.1 ± 0.1	51.8 ± 1.2	111.3 ± 1.3	130.6 ± 1.9	164.1 ± 1.2	8.5 ± 0.2	3.7 ± 0.3	21.4 ± 0.9	8.3 ± 0.7	17.3 ± 0.9
**PHBH-30ASF/10OLA**	−5.2 ± 0.2	56.3 ± 1.4	107.6 ± 2.1	127.0 ± 1.8	159.4 ± 1.6	7.4 ± 0.1	11.4 ± 0.1	26.1 ± 1.3	8.0 ± 0.8	14.7 ± 0.8
**PHBH-30ASF/20OLA**	−5.6 ± 0.3	62.6 ± 2.0	109.9 ± 2.3	127.5 ± 2.0	157.9 ± 1.5	6.4 ± 0.3	15.9 ± 0.4	22.6 ± 1.5	7.5 ± 1.2	6.7 ± 0.4

***** Δ*H_m1_* and *X_c1_* correspond to the first heating scan.

**Table 4 polymers-12-01097-t004:** Summary of the main thermal degradation parameters of PHBH-ASF/OLA composites with different compositions, in terms of onset degradation temperature (*T_2%_*), temperature of maximum degradation (*T_max_*), and residual mass at 700 °C.

Code	*T_2%_* (°C)	*T_deg_* (°C)	Residual Mass (wt %)
**ASF**	101.4 *	300.6/460.7	1.5 ± 0.2
**PHBH**	286.8	308.9	2.4 ± 0.3
**PHBH-10ASF**	253.2	288.1	2.3 ± 0.2
**PHBH-20ASF**	250.5	284.3	2.1 ± 0.1
**PHBH-30ASF**	223.6	279.1	2.0 ± 0.3
**PHBH-30ASF/10OLA**	258.4	292.0	2.0 ± 0.2
**PHBH-30ASF/20OLA**	226.3	283.5	2.0 ± 0.2

* Initial weight loss in ASF due to residual water evaporation.

**Table 5 polymers-12-01097-t005:** Main dynamic-mechanical thermal parameters of PHBH-ASF/OLA composites with different compositions: flexural storage modulus (*E’*) measured at −40 °C and 25 °C and glass transition temperature (*T_g_*), obtained by dynamic-mechanical thermal analysis (DMTA).

Code	*T_g_* (°C)	*E’* at −40 °C (MPa)	*E’* at 25 °C (MPa)
**PHBH**	10.6 ± 0.9	1869 ± 42	1345 ± 28
**PHBH-10ASF**	14.3 ± 0.8	1910 ± 49	1431 ± 40
**PHBH-20ASF**	12.0 ± 0.7	1948 ± 30	1512 ± 20
**PHBH-30ASF**	11.4 ± 0.9	2019 ± 52	1604 ± 45
**PHBH-30ASF/10OLA**	9.7 ± 0.7	1601 ± 36	1352 ± 29
**PHBH-30ASF/10OLA**	9.3 ± 0.6	853 ± 25	767 ± 23

**Table 6 polymers-12-01097-t006:** Summary of the main thermo mechanical properties of neat PHBH and PHBH-ASF/OLA with different compositions, regarding the thermal expansion, obtained by thermomechanical analysis (TMA).

Code	*T_g_* (°C)	CLTE (μm m^−1^ °C^−1^)	
		Below *T_g_*	Above *T_g_*
**PHBH**	−0.3 ± 0.1	77.1 ± 2.2	160.7 ± 2.3
**PHBH-10ASF**	0.2 ± 0.1	76.9 ± 2.1	157.0 ± 1.3
**PHBH-20ASF**	−0.4 ± 0.1	75.6 ± 2.1	157.4 ± 2.9
**PHBH-30ASF**	1.4 ± 0.2	66.8 ± 0.8	140.3 ± 2.6
**PHBH-30ASF/10OLA**	−1.3 ± 0.1	72.0 ± 0.9	169.1 ± 3.8
**PHBH-30ASF/20OLA**	−1.4 ± 0.2	90.7 ± 4.1	194.3 ± 2.83

**Table 7 polymers-12-01097-t007:** Values of the diffusion coefficient (*D*) and the corrected diffusion coefficient (*D_c_*) for PHBH and the PHBH-ASF composites processed with OLA.

Code	*D* × 10^−9^ (cm^2^ s^−1^)	*Dc* × 10^−9^ (cm^2^ s^−1^)
**PHBH**	0.14 ± 0.03	0.07 ± 0.01
**PHBH-10ASF**	0.54 ± 0.05	0.25 ± 0.02
**PHBH-20ASF**	1.56 ± 0.07	0.74 ± 0.04
**PHBH-30ASF**	6.08 ± 0.08	2.89 ± 0.05
**PHBH-30ASF/10OLA**	6.66 ± 0.09	3.17 ± 0.07
**PHBH-30ASF/20OLA**	7.08 ± 0.09	3.37 ± 0.03

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
