# Peer review of "Development and Characterization of Sustainable Composites from Bacterial Polyester Poly(3-Hydroxybutyrate-co-3-hydroxyhexanoate) and Almond Shell Flour by Reactive Extrusion with Oligomers of Lactic Acid"

_polymers, 2020, doi:10.3390/polym12051097_

Round 1
Reviewer 1 Report
“Development and characterization of sustainable composites from bacterial polyester poly(3-hydroxybutyrate-co-3-hydroxyhexanoate) and almond shell flour by reactive extrusion oligomers of lactic acid” by Ivorra-Martinez et al. details the thermal and mechanical properties of wood plastic composites in which almond shell flour is used as filler in a matrix of poly(3- hydroxybutyrate-co-3-hydroxyhexanoate). Oligomers of polylactide are used to improve the compatibility between these two components. Green composites are an important field of research and the authors present new results. They show the potential interest of this specific association. I think the paper could be published in polymers after the authors answer the following questions.
Major comments:
1) The authors regularly evoke a plasticization effect in addition to compatibilization to explain the mechanical properties when OLA is added in PHBH-ASF. This interpretation might be discussed. As shown in Figure 3 and Table 3, there is only about 5 °C difference between the Tg of PHBH-30ASF and PHBH-30ASF/20OLA, which is weak. The authors consider themselves that the effect is not spectacular on the diffusion coefficient. So could the
mechanical variations being essentially the result of the compatibilization?
Literature, if possible involving OLA, should be used to prove that such Tg difference could lead to a plasticizing effect.
2) Moreover the mechanical tests are performed about 20°C above the glass transition temperature of PHBH, i.e., when PHBH and OLA amorphous phases are in rubbery state and do not bring rigidity to the material. In this case, it is hard in my opinion to relate the modulus values to the plasticization of the amorphous phase without giving data regarding the degree of crystallinity of these materials.
It is often observed that a semi-crystalline polymer exhibits a glass transition about 5 °C higher than its amorphous homologous. In Figure 3, one can assume that the degree of crystallinity of PHBH-30ASF is higher than the one of PHBH-30ASF/20OLA. We can deduce that because the enthalpy of crystallization is lower, the glass transition is broader and
the heat capacity step is of less amplitude. This seems to indicate that PHBH-30ASF crystallized quickly during the cooling whereas the crystallization kinetic is slowed down when adding OLA. Therefore, the mechanical properties might be correlated with the degree of crystallinity.
Since the authors performed two heating scans, the first one could be used to calculate the degree of crystallinity before the erasing of the thermal history. It could be interesting to perform the calculation on the second scan also, as it gives an idea of the material propensity to recrystallize.
3) When performing the calculation, the authors should take care to the fact that the enthalpy of melting they present in Table 3 is wrong. The enthalpy of melting cannot be lower than the enthalpy of crystallization. This means that the whole melting process was not observed and that the scan should be continued to upper temperatures.
Other comments:
There are too many significant numbers in the Tables. The accuracy of thermal and mechanical tests is not so high. I suggest replacing for example 54.6±1.1 by 55±2.
Line 138: Physical aging is not an appropriate term to evoke recrystallization since it often refers to structural relaxation. Besides, I do not understand the relation between the recrystallization and the storage in desiccators. The recrystallization should essentially be linked to the temperature of storage.
Line 185: The drying at 60°C could favour the crystallization which could impact diffusion properties. Did the authors consider this point when performing their experiments?
Line 204: Equation (2) seems sufficient to calculate the coefficient of diffusion. What brings equation (3)?
Line 275: Why interactions between ester groups are not evoked as possibility for compatibilization?
Line 307: High modulus is not systematically related to the brittle character of a material. It is better to speak about rigidity or stiffness.
Line 319: The viscoelastic behaviour is intrinsic to the polymer. Please replace by liquid state or rubbery state.
Line 324: What do the authors mean by “a dilution effect”?
Line 327: “Crystallite” is more specifically used when referring to orientation processes.
Please replace by crystals.
Line 333: Discussion regarding the enthalpy of crystallization would be clearer before writing about the melting process. Here again I suggest to modify this part of the text by including the discussion about the degree of crystallinity.
Line 336: Could the authors detail the calculation (70*32.9/100)
Line 353: The authors propose that the mobility of the chains is favoured. In that case, macromolecules rearrangement should be easier and the crystallization process should also occur more easily. It is possible that OLA disturbs crystallization, essentially because it brings disorder to the structural arrangement of PHBH.
Line 359: The authors wrote that the difference regarding the temperature of melting is not significant but it is in the same order than the difference between glass transition temperatures
which is used to argue about plasticization.
Line 394: The given mass loss is the one related to the degradation of cellulose and hemicelluloses.
Figure 4a: Y-axis is not mass loss. It is mass.
Figure 4b: Please check whether it is a derivative per time or per temperature.
Line 407: What means “linear combination” there? Do the authors refer to the small hump growing with the content of ASF? This should be better explained.
Line 426: Please add when mentioning 10-30°C what refers to Tonset and what refers to Tdeg.
Line 444: same comment as line 307
Line 447: same comment as line 319
Line 455: The sentence is not clear. Could it be related to the fact that lignocellulosic components are still in their glassy state?
Lines 469-470: Plasticization is not ascertained
Line 477: The authors mention that ASF addition increases Tg. But if the content of ASF increases, Tg decreases. So, is this shift significant?
Line 496: same comment as line 307
Line 508: Here, the discussion should be revised after presenting the degree of crystallinity.
Line 511: When writing that lignocellulosic components absorb and remove water. Is that from the matrix?
Line 540: As discussed by the authors, there is no spectacular plasticizing effect on the diffusion coefficient.
Line 569: Stress concentration is not demonstrated in this study.
Typo Comments:
Line 24: why WPC with 30 wt% ASF is the most attracting material?
Line 27: phr should be defined
Lines 28-30: These two sentences could be simplified into one.
Line 33: There is no evidence of stress concentration in the text. I suggest removing this.
Line 53: please replace the by they “they represent”
Line 67: “Due to the need”
Line 75: “polylactide”
Line 78: punctuation needs to be added at the end of the sentence
Line 81 and 82: replace i by y in poly(3- hydroxybutyrate-co-3-hydroxyhexanoate)
Line 87: Please mention whether the comparison is made with P3HB
Line 101: Please replace by “to increase toughness”
Line 108: Please replace by “so requires an appropriate temperature profile”
Line 114: “ester content”
Line 126: phr should be defined
Line 174: please replace characterization by analysis
Line 198: can be done
Line 223: please replace by Singh et al. established that the decrease of tensile strength results
from stress concentrations at the polymer/filler interfaces.
Table2: Please replace Strenght by Strength
Line 260: The sentence can be removed. It repeats what has been said above. Moreover, I
suggest moving the following sentence line 262 to the discussion line 229 about the lack of
interface.
Line 265: Please replace this lack by the lack
Line 267: This sentence can also be removed. Instead it is possible to end the previous
sentence by “and the stress transfer”
Line 276: Where is the white arrow on the picture?
Line 276: Please replace improved by improve
Line 348: Please replace “than” by “in comparison with”
Line 358: Please replace by “a much lower Dcc”
Line 372: A segment seems being repeated.
Line 377: Please replace similar same by similar.
Line 377: After “the degradation process”
Line 389: What means “two amin steps”?
Line 442: Please replace “ASF wt% and the presence or not of OLA” by “ASF and OLA wt%”
Line 451: observe for
Line 474: What is the meaning of “to most widely extended”? “the most used”?
Line 489: “is favoured”
Line 490: Please replace “of PHBH-ASF/OLA composites” by “of neat PHBH and
composites”
Line 498: A parenthesis is missing.
Line 499: “thus leading to”
Lines 528-533: There are several repetitions in these sentences that could be simplified.
Line 544: “think on” instead of “thing on”
Line 550: ASF instead of ASD
Line 555: Please delete “additions”
Author Response
Changes according Reviewer 1
Major comments:
1) The authors regularly evoke a plasticization effect in addition to compatibilization to explain the mechanical properties when OLA is added in PHBH-ASF. This interpretation might be discussed. As shown in Figure 3 and Table 3, there is only about 5 °C difference between the Tg of PHBH-30ASF and PHBH-30ASF/20OLA, which is weak. The authors consider themselves that the effect is not spectacular on the diffusion coefficient. So could the
mechanical variations being essentially the result of the compatibilization?
Literature, if possible involving OLA, should be used to prove that such Tg difference could lead to a plasticizing effect.
ANSWER
We agree with the reviewer comment. It is true that oligomers of lactic acid (OLA), are usually added to PLA to provide a remarkable plasticization effect; but it is also true that the plasticization efficiency highly depends on the PLA grade and on the OLA grade (molecular weight, branching, viscosity, and so on). Therefore, we have provided a new paragraph showing the wide range of plasticization effects that OLA can provide to PLA and PLA/PHA blends to show this disparity. We have not found an in depth study of the plasticization effect of OLA on PHBH, but the particular structure of the PHBH, with medium-length 3-hydroxyhexanoate, could be a restriction to remarkable decrease in Tg. To clarify this, in accordance to the reviewer comment, we have added a paragraph in text and secondary literature to support this.
“The low molecular weight OLA chains offer slightly increased mobility in comparison with the polymer chains of neat PHBH, similar results were reported by Quiles-Carrillo et al. [55] . This phenomenon increases the free volume in the polymeric structure and leads to a poor plasticizing effect. It is true that addition of oligomers of lactic acid to PLA, usually leads to a remarkable decrease in PLA’s Tg as reported by Burgos et al. [56] from 66 °C (neat PLA) down to -10 °C. Nevertheless, the decrease in Tg, is directly related to the PLA and OLA structure. D. Lascano et al. [50] reported a decrease of PLA Tg from 63.3 °C (neat PLA; different grade) down to 50.8 °C with 20% OLA (different commercial grade of OLA). Armentano et al. [57] reported a dual plasticization effect of PHB and OLA on PLA, but the decrease in Tg was not as important as the above-mentioned by Burgos et al. Moreover, Amor et al. [58], reported a slight plasticization effect on PLA/PHBH blends by using OLA in pellet form with a dual plasticization effect of PHBH (which provided a 3 °C decrease in Tg of PLA with a PHBH loading of 10 wt%) and OLA (which provided a decrease of 1 °C with a load of 1 wt%. All these results show the disparity in plasticization of PLA with OLA even they share the same chemical structure. Therefore, the plasticization effect on PHAs is even more complex and has not been studied previously independently of blends with PLA. In this work, this decrease in Tg values are very low, but we must bear in mind that PHBH structure contains medium chain hydroxyalkanoates and these, contribute to lowering crystallinity compared to P3HB. Therefore, this slight decrease could representative of somewhat plasticization effect provided by OLA.”
2) Moreover the mechanical tests are performed about 20°C above the glass transition temperature of PHBH, i.e., when PHBH and OLA amorphous phases are in rubbery state and do not bring rigidity to the material. In this case, it is hard in my opinion to relate the modulus values to the plasticization of the amorphous phase without giving data regarding the degree of crystallinity of these materials.
It is often observed that a semi-crystalline polymer exhibits a glass transition about 5 °C higher than its amorphous homologous. In Figure 3, one can assume that the degree of crystallinity of PHBH-30ASF is higher than the one of PHBH-30ASF/20OLA. We can deduce that because the enthalpy of crystallization is lower, the glass transition is broader and
the heat capacity step is of less amplitude. This seems to indicate that PHBH-30ASF crystallized quickly during the cooling whereas the crystallization kinetic is slowed down when adding OLA. Therefore, the mechanical properties might be correlated with the degree of crystallinity.
Since the authors performed two heating scans, the first one could be used to calculate the degree of crystallinity before the erasing of the thermal history. It could be interesting to perform the calculation on the second scan also, as it gives an idea of the material propensity to recrystallize.
ANSWER
Following the recommendations of the reviewer, we have gone back to the raw DSC thermograms and we have completed the thermal analysis characterization. A third melting hump at 162 ºC has been also considered for calculating the melting enthalpy of PHBH (we have already added secondary literature to support PHBH melting process in three overlapped peaks).
|
Code |
Tg (ᵒC) |
Tcc (ᵒC) |
Tm1 (ᵒC) |
Tm2 (ᵒC) |
Tm3 (ᵒC) |
ΔHm1 (J g-1) |
ΔHcc2 (J g-1) |
ΔHm2 (J g-1) |
Xc1 (%) |
Xc2 (%) |
|
PHBH |
0.3±0.1 |
54.6±1.1 |
111.5±1.9 |
130.8±2.0 |
162.5±1.2 |
20.3±0.5 |
26.7±0.8 |
33.6±1.3 |
13.9±1.1 |
4.7±0.3 |
|
PHBH-10ASF |
-0.5±0.1 |
57.5±1.8 |
112.1±1.8 |
129.7±1.7 |
162.3±1.8 |
19.6±0.4 |
17.4±0.4 |
29.0±2.2 |
14.9±1.1 |
8.8±0.4 |
|
PHBH-20ASF |
-1.9±0.2 |
55.5±1.9 |
111.5±2.0 |
129.6±2.1 |
163.4±1.4 |
18.1±0.3 |
15.0±0.1 |
26.4±2.1 |
15.5±0.8 |
9.8±0.7 |
|
PHBH-30ASF |
-1.1±0.1 |
51.8±1.2 |
111.3±1.3 |
130.6±1.9 |
164.1±1.2 |
8.5±0.2 |
3.7±0.3 |
21.4±0.9 |
8.3±0.7 |
17.3±0.9 |
|
PHBH-30ASF/10OLA |
-5.2±0.2 |
56.3±1.4 |
107.6±2.1 |
127.0±1.8 |
159.4±1.6 |
8.2±0.1 |
11.4±0.1 |
26.12±1.3 |
8.8±0.8 |
14.7±0.8 |
|
PHBH-30ASF/20OLA |
-5.6±0.3 |
62.6±2.0 |
109.9±2.3 |
127.5±2.0 |
157.9±1.5 |
6.4±0.3 |
15.9±0.4 |
22.6±1.5 |
7.5±1.2 |
6.7±0.4 |
“The addition of a filler takes effect on the degree of crystallinity. During the first heating scan PHBH reaches a Xc value of 13.9%. This increases with the amount of ASF filler up to 15.5% with 20 wt% ASF which suggests that ASF (mainly crystalline cellulose fractions) acts as a nucleant agent [61]. Furthermore with the addition of 30 wt% ASF, Xc1 decreases to 8.3% due to the decrease of free volume necessary for nucleation of polymer, since high lignocellulosic filler content can hinder crystal formation [62]. Mechanical characterization shows no correlation between the degree of crystallinity, while Elastic Modulus increases up to 30 wt% ASF, the degree of crystallinity is saturated with only 20 wt%. The degree of crystallinity is different in the second heating scan because the recrystallization conditions are different. While on the first heating scan PHBH has been aged at 25 °C for 15 days, on the second heating scan the materials were recrystallized under controlled conditions above-mentioned. On the second heating scan, the same tendency is observed until 30 wt% ASF. The compatibilizing effect of OLA could negatively contribute increase the degree of crystallinity by reducing the gaps between the filler and the matrix as it is reported in FESEM analysis [63].”
3) When performing the calculation, the authors should take care to the fact that the enthalpy of melting they present in Table 3 is wrong. The enthalpy of melting cannot be lower than the enthalpy of crystallization. This means that the whole melting process was not observed and that the scan should be continued to upper temperatures.
ANSWER
As indicated previously, we have gone back to the raw data regarding DSC analysis and we have found (as indicated in the new Figure 3), we missed the evaluation of the melt enthalpy corresponding to a third melt peak located at 162 ºC. It is evident, the melting process of PHBH is quite complex due to different crystallites and/or potential meting and recrystallization. This leads to a melt process that covers the 100 – 164 ºC. Accordingly, to these DSC graphs, all thermal parameters have been recalculated and provided in the extended Table 3. In this table DHm1, stands for the sum of all three melting peaks and, consequently, as indicated by the reviewer, the new obtained results are show consistency.
Other comments:
There are too many significant numbers in the Tables. The accuracy of thermal and mechanical tests is not so high. I suggest replacing for example 54.6±1.1 by 55±2.
ANSWER
We usually use three significative figures to represent the results, but as the reviewer suggests, some techniques are not high accurate and, although they can provide a series of infinite decimal numbers, it is coherent to round them to a reasonable value and standard deviation. Therefore, we have followed the reviewer’s recommendation for some of the mechanical and thermal parameters. In addition, the values in table 5 have been modified to reduce the number of significant numbers.
Line 138: Physical aging is not an appropriate term to evoke recrystallization since it often refers to structural relaxation. Besides, I do not understand the relation between the recrystallization and the storage in desiccators. The recrystallization should essentially be linked to the temperature of storage.
ANSWER
We are in total agreement with the reviewer’s comment about aging; it is better use the term secondary crystallization but sometimes this is called an aging process since the mechanical performance of the processed materials are remarkable changes (usually an embrittlement) occur during this secondary crystallization. It is true that the recrystallization is directly related to the temperature. Therefore, we used 25 ºC to represent a recrystallization process at room temperatures. The use of a vacuum desiccator was only to provide a constant relative humidity due to the sensitiveness of polyesters to hydrolysis.
“It is well known that bacterial polyesters undergo secondary crystallization or recrystallization with time (sometimes designed as physical aging since this leads to an embrittlement), especially at temperatures above Tg. Recrystallization rate is directly related to temperature; therefore samples have been subjected to a recrystallization process at 25 °C for 15 days since it has been reported that almost all recrystallization takes place after two weeks from the processing [17,19,20]. To avoid potential hydrolysis of the polyester surface, samples were stored in a vacuum desiccator with constant moisture.”
Line 185: The drying at 60°C could favour the crystallization which could impact diffusion properties. Did the authors consider this point when performing their experiments?
ANSWER
We thank reviewer for this question. It is true that using a drying process at 60 ºC for 24h could induce crystallization. We did not take this into account in water uptake experiment, since all the materials considered have the same thermal treatment. The main reason for doing that was to remove a much moisture from the sample. A future work could be focused on the effect of an annealing process on diffusion properties of PHBH.
Line 204: Equation (2) seems sufficient to calculate the coefficient of diffusion. What brings equation (3)?
ANSWER
We agree with the reviewer. Equation 3 is unnecessary as it can be deduced by simple mathematical operations on Equation 2. Therefore, it has been removed as it does not give any additional information. As other expressions have been added, the numbering is different, but the previous Equation 3 has been removed as it can be seen in the revised version of the manuscript.
Line 275: Why interactions between ester groups are not evoked as possibility for compatibilization?
ANSWER
As recommended by the reviewer, an additional paragraph related to the potencial interactions between ester groups and cellulose particles has been included in the revised version. In addition, new secondary literature has been provided to support the new statements.
“Despite high polarity ester groups can establish somewhat interactions with polar groups in cellulose, the main compatibilizing effects are obtained with high reactive groups such as maleic anhydride, carboxyl acids, end-chain hydroxyl groups, glycidyl methacrylate as reported by Pracela et al. [45] by using functionalized copolymers to provide increased interface interactions between a polymer matrix and cellulose particles. Some interactions between ester groups and cellulose particles have been described by Chabros et al. [46] in thermosetting unsaturated polyester resins with cellulose fillers; in particular they describe some interactions between the polar ester groups and hydroxyl groups in cellulose by hydrogen bonding. These small range interactions can also occur in PHBH/ASF composites, but their intensity is lower than that provided by the reaction of carboxylic acid and hydroxyl terminal groups in OLA with both hydroxyl groups in cellulose and PHBH through condensation or esterification reactions. As described by Mokhena et al. [47] the ester groups in PLA are not enough to provide intense interactions between the polyester-type matrix and the cellulose filler. They report the need of different treatments on cellulose such as acetylation, glyoxalization, silylation, treatment with glycidyl methacrylate (GMA), among others to improve polymer-matrix interactions. This could be related not only to the polarity but also with the hydrophilic nature of ASF and the hydrophobic nature of PHBH.”
Line 307: High modulus is not systematically related to the brittle character of a material. It is better to speak about rigidity or stiffness.
ANSWER
We agree with this comment. Thank you for the recommendation. We have corrected in the revised version.
Line 319: The viscoelastic behaviour is intrinsic to the polymer. Please replace by liquid state or rubbery state.
ANSWER
We agree with this comment. It has been corrected.
Line 324: What do the authors mean by “a dilution effect”?
ANSWER
Thanks to the reviewer for asking this. The term dilution mans the following. If we observe the DSC of neat PHBH, the cold crystallization process appears located at 50 ºC and its heigh is the maximum for this polymer and in these conditions. Nevertheless, PHBH composites with 30 wt% ASF, the peak related to the PHBH cold crystallization, takes place almost at the same temperature but its height has been reduced because the sample contains 30 wt% ASF which has thermal transition in this temperature range. Maybe it was not clearly explained and, therefore, we have added a sentence to clarify this and avoid misunderstanding.
“The addition of lignocellulosic ASF particles does not influence the Tcc of PHBH in the developed composites. It can only be observed a dilution effect (the cold crystallization peak height is smaller in PHBH/ASF composites, compared to neat PHBH; this is because the PHBH/ASF composite contains, e.g. 30 wt% ASF which has no thermal transition in this temperature range, and consequently, the intensity of the peak is lower).”
Line 327: “Crystallite” is more specifically used when referring to orientation processes.
Please replace by crystals.
ANSWER
Corrected.
Line 333: Discussion regarding the enthalpy of crystallization would be clearer before writing about the melting process. Here again I suggest to modify this part of the text by including the discussion about the degree of crystallinity.
ANSWER
As indicated in a previous question, we have gone back to the original DSC curves for all three stages and we have analyzed and calculated new thermal parameters from DSC runs, which are more representative for the actual thermal behaviour of these composites. In addition, the degree of crystallinity has been calculated and related to some other thermal parameters.
Line 336: Could the authors detail the calculation (70*32.9/100)
ANSWER
Thank you for the question. This is just a theoretical calculation that would be real if neat PHBH and the PHBH in PHBH/30 wt% ASF composite had the same cold crystallization enthalpy. Then, the dilution effect will give a diluted enthalpy for the composite with 30 wt% ASF of 0.7*32.9, which gives 23.03 J/g, directly by integration of the cold crystallization peak, since the DSC curve does not take into account the composition to calculate the enthalpy. After integration of the cold crystallization peak, the result is the energy per unit mass (1 g), but in the composite, the only component that contributes to the exothermic cold crystallization peak is PHBH; then the unit mass corresponds to (0.7 g PHBH+0.3 g ASF). Therefore, this number, 23.03 J/g, will represent the enthalpy that the software would give by considering the dilution effect. But as one can observe, the obtained enthalpy is much lower, which could indicate ASF particles, at high loading, hinder crystallization.
As we have recalculated all thermal parameters and values, it is possible to observe the dilution effect on melt enthalpies, but this tendency is not observed in cold crystallization enthalpies as indicated in this table. As one can see, the dilution effect means multiplying the maximum enthalpy by 1, 0.9, 0.8 or 0.7 for 0, 10, 20 and 30 wt% PHBH respectively. The results corresponding to the dilution effect regarding the melt enthalpy, are almost coincident, but the results corresponding to the dilution effect of the cold crystallization enthalpy are clearly different. Therefore, ASF particles affect the cold crystallization process.
|
Code |
wt% ASF |
ΔHcc2 (J g-1) |
ΔHcc2 (J g-1)
DILUTION EFFECT |
ΔHm2 (J g-1) |
ΔHcc2 (J g-1)
DILUTION EFFECT |
|
PHBH |
0 |
26.7±0.8 |
26.7 |
33.6±1.3 |
33.6 |
|
PHBH-10ASF |
10 |
17.4±0.4 |
24.03 |
29.0±2.2 |
30.24 |
|
PHBH-20ASF |
20 |
15.0±0.1 |
21.36 |
26.4±2.1 |
26.88 |
|
PHBH-30ASF |
30 |
3.7±0.3 |
18.69 |
21.4±0.9 |
23.52 |
To clarify this, the following paragraph has been added to the revised version to give more clarity about the dilution effect and the enthalpy values.
“The addition of lignocellulosic ASF particles does not produce significant changes in melting temperatures as observed in other studies [34]. The analysis of the enthalpies corresponding to the cold crystallization process (ΔHcc), indicated that the highest enthalpy corresponded to neat PHBH during the second heating cycle (ΔHcc2 = 26.7 J g-1). These values decreased gradually with the addition of ASF particles, down to values of 3.7 J g-1 for the sample with 30 wt% ASF. The dilution effect (which means considering the actual PHBH content without taking into account the wt% ASF for the cold crystallization enthalpy calculation), would give a theoretical diluted enthalpy of 18.69 J g-1, which is remarkably higher than the actual obtained value of 3.7 J g-1. These differences are not so pronounced for 10 and 20 wt% ASF; therefore, these results suggest high wt% ASF particles hinder PHBH cold crystallization.”
Line 353: The authors propose that the mobility of the chains is favoured. In that case, macromolecules rearrangement should be easier and the crystallization process should also occur more easily. It is possible that OLA disturbs crystallization, essentially because it brings disorder to the structural arrangement of PHBH.
ANSWER
We agree with the reviewer. In fact, the shift is almost 10 ºC for 20 phr OLA typical plasticizers move the cold crystallization to lower temperatures due to increased chain mobility. But in this case, OLA does not provide that effect. In contrast, addition of OLA delays the cold crystallization process. We have searched for secondary literature to rewrite this and we have found that some plasticizer/compatibilizers react with the polymer to form a branched (and sometimes, some cross-linked) structures that break uniformity and, subsequently, delay the cold crystallization process. This paragraph has been added to the revised version, which a more clear explanation on the potential effects of OLA.
“This is not the typical effect of a plasticizer which increases chain mobility and, therefore, the cold crystallization process is shifted to lower temperatures as reported by Lascano et al. [50] and Ferri et al. [59]. Nevertheless, some additives such as maleinized linseed oil (MLO), maleinized cottonseed oil (MCSO), or even epoxidized vegetable oils, promote an overlapping of several phenomena such as slight plasticization, chain extension, branching and, in some cases, potential crosslinking due to the multifunctionality of these compounds. Some of these vegetable-oil derivatives, produce the same behaviours as observed in this study, i.e. a shift of the cold crystallization process to higher temperatures, due to the disrupted overall structure they provide with branches, chain extension, and so on as reported by Garcia-Campo et al. [60]. Liminana et al. [61] reported the potential of these modified vegetable oils as compatibilizers for PBS and lignocellulosic fillers, due to reaction of oxirane, maleic anhydride groups with hydroxyl groups in almond shell flour. In this study, it seems OLA has a similar effect to those modified vegetable oils, since changes in Tg are very low (which is representative for very slight plasticization effect, almost inexistent, since the technique itself has some uncertainty in the obtained values depending on the sample size, geometry, surface contact, and so on), and the cold crystallization is shifted to higher temperatures, thus indicating that other phenomena could be occurring, such as compatibilization and/or chain extension.”
Line 359: The authors wrote that the difference regarding the temperature of melting is not significant but it is in the same order than the difference between glass transition temperatures which is used to argue about plasticization.
ANSWER
After carefully reading the manuscript, we agree with the reviewer. Changes in Tg are only of 5 ºC and changes in melt peak temperature (2, 3 and 4 ºC for Tm1, Tm2 and Tm2, respectively). It is true that the slight changes in Tg are representative for very poor plasticization effects as stated in the manuscript. Even more, these changes are low enough to fall within the technique error itself. Therefore, these slight changes cannot be considered in both processes, the glass transition process and the melting. We have rewritten some sentences to avoid confusion.
Line 394: The given mass loss is the one related to the degradation of cellulose and hemicelluloses.
ANSWER
As indicated by the reviewer, this was a mistake. Subsequently it has been corrected with the corresponding weight % loss, attributable to cellulose and hemicellulose degradation.
“The first step of weight loss, of about 44.2 wt%, corresponds to the degradation of the cellulose and hemicellulose contained in ASF particles. The first component to start degradation is hemicellulose, followed by cellulose and lignin. Lignin shows a slower (in a wide temperature range) degradation process, so the third section of the TGA curve shows a lower slope, starting at 357.3 °C (temperature change of slope in the thermogram) up to 500 °C, with a loss of 47.6 wt%.”
Figure 4a: Y-axis is not mass loss. It is mass.
ANSWER
Thanks for the recommendation. We have changed it in the revised version.
Figure 4b: Please check whether it is a derivative per time or per temperature.
ANSWER
It is an interesting question. We have gone back to the raw TGA data, and the calculated DTG is per unit time.
Line 407: What means “linear combination” there? Do the authors refer to the small hump growing with the content of ASF? This should be better explained.
ANSWER
In order to clarify this, and considering the reviewer’s recommendation, we have made a modification for a better understanding
“The TGA curves of PHBH-ASF composites indicated that the thermal degradation process is a linear combination of the two individual degradation phenomena observed in PHBH and ASF, the first part of the curve is identical (or very similar) neat PHBH and, a small hump at the end of the curve can be detected, which is attributable to residual degradation of ASF which changes with the ASF wt%.”
Line 426: Please add when mentioning 10-30°C what refers to Tonset and what refers to Tdeg.
ANSWER
As recommended by the reviewer, we have made a modification for a better understanding
“The addition of 10 phr OLA in small amounts seems to slightly improve the thermal stability of the developed composites. The characteristic thermal degradation values are delayed by 34 °C (onset degradation temperature, T2%) and by 10 °C (for the maximum degradation rate, Tdeg) compared to the unmodified reference composite without OLA.“
Line 444: same comment as line 307
ANSWER
Thanks for the comment. We have changed it according the comment on line 307.
Line 447: same comment as line 319
ANSWER
Thanks for the comment. We have changed it according to the comment in line 319.
Line 455: The sentence is not clear. Could it be related to the fact that lignocellulosic components are still in their glassy state?
ANSWER
We agree with the reviewer. In this temperature range, lignocellulosic components are below their Tg and, consequently they offer a glassy behaviour which contributes to increased stiffness. Nevertheless, the PHBH matrix represents the major component and, overall properties, are governed by its behaviour. To give some additional clarity to this statement, this issue has been commented.
“In addition, at this temperature range, lignocellulosic components are below its Tg, which means they show a glassy behaviour that promotes increased stiffness. Nevertheless, it is important to bear in mind that the main component in PHBH/ASF composites is PHBH and the dynamic behaviour is highly influenced by PHBH behaviour since conditions are not as aggressive as a conventional tensile test up to fracture.”
Lines 469-470: Plasticization is not ascertained
ANSWER
Accordingly to other comments of the reviewer, we have changed all sentences related to the plasticization effect, since it seems OLA has a positive effect on mechanical ductile properties, but the decrease in Tg, typical of good plasticization is very small in DSC, and even more smaller by DMTA. Regarding this paragraph, it has been rewritten as:
“It is worthy to note the extremely small changes in Tg obtained by DMTA which suggests, as DSC, very poor plasticization effect, suggesting compatibilization one of the most representative effects of OLA in this PHBH/ASF system.”
Line 477: The authors mention that ASF addition increases Tg. But if the content of ASF increases, Tg decreases. So, is this shift significant?
ANSWER
By considering different comments of the reviewer about the plasticization effect, we have checked all references to plasticization in the text and avoided sentences like “a slight plasticization effect can be seen due to 2 ºC decrease”, since the changes in Tg values obtained by the different techniques are not significant, and cannot be used to support plasticization. Accordingly to this, regarding DMTA characterization, the paragraph has been rewritten as:
“It can be observed that tan d peaks are slightly moved towards higher temperatures in uncompatibilized PHBH-ASF composites, compared to neat PHBH (a maximum shift of 3 – 4 °C). This change is not significant as observed in other techniques such as DSC. On the other hand, the addition of OLA leads to slightly lower Tg values by DMTA, but once again, these changes are not high enough to give a clear evidence of the chain mobility restriction by ASF particles or increased chain mobility by OLA.”
Line 496: same comment as line 307
ANSWER
Thanks for the comment. We have changed it according the comment on line 307.
Line 508: Here, the discussion should be revised after presenting the degree of crystallinity.
ANSWER
As recommended by the reviewer, and with the additional information about the degree of crystallinity, the discussion of this section has been carried out by taking into account both variables, i.e. wt% lignocellulosic filler and degree of crystallinity of the matrix polymer (%). This is the new paragraph included in the revised version.
“Obviously, due to the hydrophobicity of neat PHBH, it shows the lowest water absorption for 9 weeks. After 35 days of immersion it reaches a constant saturation mass, ∆mass∞ of 0.53%, which is maintained practically until 63 days of immersion. As above-mentioned, the Xc of neat PHBH is 13.9%. The addition of ASF (highly hydrophilic particles due to its composition: cellulose, hemicellulose and lignin as the main components) considerably increases water absorption, but the effects of PHBH crystallinity can also be observed in this behaviour. The composite with 10 wt% ASF reaches a water saturation mass, ∆mass∞ of 1.46 wt% after 42 days (Xc of PHBH in this composite is 14.9% and this prevents from water entering). In a similar way, uncompatibilized composites containing 20 wt% show a relatively low ∆mass∞ of 3.1 wt%. This is almost double the previous value (with 10 wt% ASF). This value is expected since the Xc of PHBH in this composite is close to 15.5%. Nevertheless, the composite containing 30 wt% ASF, shows a remarkable increase in ∆mass∞ up to values of 7.1 wt% after 9 weeks, which represents almost 14 time the value for neat PHBH, representing a typical result in most WPCs [23,36,42]. The result is higher than the extrapolation from the ASF wt% (should be of about three times the value of the composite with 10 wt% ASF, i.e. 4.5 wt%); despite this, we have to bear in mind that the Xc of PHBH in this composite has decreased down to 8.3% and this has a negative effect on water absorption as amorphous regions allow water entering [69,70]. Therefore, the increase in water absorption is not only related to the amount of lignocellulosic components, but also with the degree of crystallinity of the polymer matrix. Cellulose, promotes water absorption due to hydroxyl (-OH) groups that interact with water molecules [21-23,25,40,71].”
Line 511: When writing that lignocellulosic components absorb and remove water. Is that from the matrix?
ANSWER
We agree with the reviewer, this concept is not clear. Therefore, we have added a new paragraph with a clear explanation on this phenomenon.
“The water absorption capacity of WPCs is an important feature in some applications due to the lignocellulosic component. This creates a three-dimensional path inside the polymer matrix that allows water entering (for example when the composite is subjected to high relative humidity environments) and this causes an expansion. It is possible that after this initial stage, this WPC could be subjected to drying at sun with low humidity; then this 3D-path allows water/moisture removal, promoting a contraction. This situation is quite usual in WPCs such as those used in fences, decking, and so on. This repeated expansion-contraction cycles could lead to formation of microcracks.”
Line 540: As discussed by the authors, there is no spectacular plasticizing effect on the diffusion coefficient.
ANSWER
As indicated by the reviewer regarding discussion of section “3.4 Evolution of the water uptake and water diffusion process in PHBH-ASF/OLA composites.”, by considering the degree of crystallinity. The same tendency can be observed for the diffusion coefficients, since D is clearly related to both parameters, i.e. wt% lignocellulosic content and degree of crystallinity. The comments about this have been rewritten. And the very slight changes in D values for composites containing 10 and 20 phr OLA, are representative for a very poor plasticizing effect as the free volume does not increase.
“Thus, it was deduced that the amount of wt% ASF is the parameter with the greatest influence on the water diffusion process in PHBH-ASF/OLA composites, together with the degree of crystallinity as previously discussed with the relationship of the water absorption and the wt% ASF loading and the degree of crystallinity of the PHBH matrix. Since the Dc values for the composites with 10 and 20 phr OLA are similar to that of the same composite with 30 wt% ASF, it is possible to conclude the poor plasticization effect of this OLA, suggesting, once again, that compatibilization is the main acting mechanisms of OLA on PHBH/ASF composites.”
Line 569: Stress concentration is not demonstrated in this study.
ANSWER
As recommended by the reviewer, since we have not shown any evidence of the stress concentration phenomenon, we have removed this sentence in the revised version.
Typo Comments:
Line 24: why WPC with 30 wt% ASF is the most attracting material?
ANSWER
As indicated by the reviewer, we have given some additional information on why the composite with 30 wt% ASF is an interesting option from an industrial point of view.
“Nevertheless, for real applications, the WPC with 30 wt% ASF is the most attracting material since it contributes to lowering the overall cost of the WPC and can be manufactured by injection moulding, but its properties are really compromised due to the lack of compatibility between the hydrophobic PHBH matrix and the hydrophilic lignocellulosic filler.”
Line 27: phr should be defined
ANSWER
Thanks for your comment. We have defined it.
“To minimize this phenomenon, 10 and 20 phr (weight parts of OLA per one hundred weight parts of PHBH) were added to PHBH/ASF (30 wt% ASF) composites”
Lines 28-30: These two sentences could be simplified into one.
ANSWER
As recommended by the reviewer, these sentences have been rewritten.
“Differential scanning calorimetry (DSC) suggested poor plasticization effect of OLA on PHBH-ASF composites. Nevertheless, the most important property OLA can provide to PHBH/ASF composites is somewhat compatibilization since some mechanical ductile properties are improved with OLA addition.“
Line 33: There is no evidence of stress concentration in the text. I suggest removing this.
ANSWER
As suggested, it has been removed.
Line 53: please replace the by they “they represent”
ANSWER
Corrected.
Line 67: “Due to the need”
ANSWER
Corrected.
Line 75: “polylactide”
ANSWER
Corrected.
Line 78: punctuation needs to be added at the end of the sentence
ANSWER
Corrected.
Line 81 and 82: replace i by y in poly(3- hydroxybutyrate-co-3-hydroxyhexanoate)
ANSWER
Corrected.
Line 87: Please mention whether the comparison is made with P3HB
ANSWER
We have carefully read these references and the refer to other studies in which, a comparison of copolymers with P3HB is carried out by using different techniques.
- Yang, Y.; Ke, S.; Ren, L.; Wang, Y.; Li, Y.; Huang, H. Dielectric spectroscopy of biodegradable poly (3-hydroxybutyrate-co-3-hydroxyhexanoate) films. European polymer journal 2012, 48, 79-85.
- Liao, Q.; Noda, I.; Frank, C.W. Melt viscoelasticity of biodegradable poly (3-hydroxybutyrate-co-3-hydroxyhexanoate) copolymers. Polymer 2009, 50, 6139-6148.
Line 101: Please replace by “to increase toughness”
ANSWER
Corrected.
Line 108: Please replace by “so requires an appropriate temperature profile”
ANSWER
Corrected.
Line 114: “ester content”
ANSWER
Corrected.
Line 126: phr should be defined
ANSWER
Thanks for the recommendation. We have defined it.
Line 174: please replace characterization by analysis
ANSWER
Thank you for the recommendation. We have defined it.
Line 198: can be done
ANSWER
Corrected.
Line 223: please replace by Singh et al. established that the decrease of tensile strength results
from stress concentrations at the polymer/filler interfaces.
ANSWER
As recommended, it has been changed.
Table2: Please replace Strenght by Strength
ANSWER
Corrected.
Line 260: The sentence can be removed. It repeats what has been said above. Moreover, I
suggest moving the following sentence line 262 to the discussion line 229 about the lack of
interface.
ANSWER
Following the recommendations of the reviewer, we have moved this sentence (260-262) to the end of line 229.
Line 265: Please replace this lack by the lack
ANSWER
Corrected.
Line 267: This sentence can also be removed. Instead it is possible to end the previous
sentence by “and the stress transfer”
ANSWER
The sentence has been removed and the previous sentence has been finished with the recommended words.
Line 276: Where is the white arrow on the picture?
ANSWER
Thank you very much for your comment. We have added the white arrows referred in the text.
Line 276: Please replace improved by improve
ANSWER
Corrected.
Line 348: Please replace “than” by “in comparison with”
ANSWER
Corrected.
Line 358: Please replace by “a much lower Dcc”
ANSWER
Corrected.
Line 372: A segment seems being repeated.
ANSWER
Corrected.
Line 377: Please replace similar same by similar.
ANSWER
Corrected.
Line 377: After “the degradation process”
ANSWER
Corrected to “The Endset of the degradation process”.
Line 389: What means “two amin steps”?
ANSWER
Corrected. It was a typo. It should have been “main”.
Line 442: Please replace “ASF wt% and the presence or not of OLA” by “ASF and OLA wt%”
ANSWER
Corrected.
Line 451: observe for
ANSWER
Corrected.
Line 474: What is the meaning of “to most widely extended”? “the most used”?
ANSWER
According to reviewer’s suggestion, we have changed the term to “the most used”.
Line 489: “is favoured”
ANSWER
Corrected.
Line 490: Please replace “of PHBH-ASF/OLA composites” by “of neat PHBH and
composites”
ANSWER
Replaced.
Line 498: A parenthesis is missing.
ANSWER
Parenthesis added at the end.
Line 499: “thus leading to”
ANSWER
Corrected.
Lines 528-533: There are several repetitions in these sentences that could be simplified.
ANSWER
As suggested in a previous question, this paragraph has been completely rewritten considering both the influence of the lignocellulosic filler and the degree of crystallinity of the PHBH matrix. Therefore, repetitions have disappeared.
“Obviously, due to the hydrophobicity of neat PHBH, it shows the lowest water absorption for 9 weeks. After 35 days of immersion it reaches a constant saturation mass, ∆mass∞ of 0.53%, which is maintained practically until 63 days of immersion. As above-mentioned, the Xc of neat PHBH is 13.9%. The addition of ASF (highly hydrophilic particles due to its composition: cellulose, hemicellulose and lignin as the main components) considerably increases water absorption, but the effects of PHBH crystallinity can also be observed in this behaviour. The composite with 10 wt% ASF reaches a water saturation mass, ∆mass∞ of 1.46 wt% after 42 days (Xc of PHBH in this composite is 14.9% and this prevents from water entering). In a similar way, uncompatibilized composites containing 20 wt% show a relatively low ∆mass∞ of 3.1 wt%. This is almost double the previous value (with 10 wt% ASF). This value is expected since the Xc of PHBH in this composite is close to 15.5%. Nevertheless, the composite containing 30 wt% ASF, shows a remarkable increase in ∆mass∞ up to values of 7.1 wt% after 9 weeks, which represents almost 14 time the value for neat PHBH, representing a typical result in most WPCs [23,36,42]. The result is higher than the extrapolation from the ASF wt% (should be of about three times the value of the composite with 10 wt% ASF, i.e. 4.5 wt%); despite this, we have to bear in mind that the Xc of PHBH in this composite has decreased down to 8.3% and this has a negative effect on water absorption as amorphous regions allow water entering [69,70]. Therefore, the increase in water absorption is not only related to the amount of lignocellulosic components, but also with the degree of crystallinity of the polymer matrix. Cellulose, promotes water absorption due to hydroxyl (-OH) groups that interact with water molecules [21-23,25,40,71].”
Line 544: “think on” instead of “thing on”
ANSWER
Corrected.
Line 550: ASF instead of ASD
ANSWER
Corrected.
Line 555: Please delete “additions”
ANSWER
Deleted.

Reviewer 2 Report
This manuscript deals with the properties of PHBH-ASF composites as well as those added with oligolactic acid (OLA). The authors carefully conducted their experiments, and the data shown here may be reliable. I was however revealed that the properties of these composites are not so highly improved as expected probably because the interfacial interaction between the matrix polymer and fillers is weak. The control of ductility (toughness) and hardness (strength) is in a trade-off relation with the amount of fillers and plasticizer. Therefore, the present manuscript must be revised by considering the following points prior to its publication.
- The title: “reactive extrusion oligomers” Why is this phrase necessary? Its explanation should be done somewhere in the text.
- 1 Materials. The molecular weight and unit composition of PHBH (i.e., 3HH unit ratio) must be shown. The degree of polymerization of OLA should also be shown here. These are fundamental characteristics of polymers. The size distribution of ASF must be shown. It can be calculated from Figure 1.
- Figure 2. No arrow is shown although it is explained in line 276.
- 1 Mechanical properties……. As mentioned above, the present composites seem to have less improved mechanical properties even when OLA is added as a compatibilizer. In this regard, the authors are requested to show the target values of mechanical properties that are needed for their real application as WPC. Then, it should be discussed how well the present composites justify the indices or how highly balanced properties they have. Without this discussion, no one understands whether the present composites are really versatile or not.
- Figure 3 and Table 3. The heat of crystallization and heat of crystal fusion are not correspondent to each other. The latter value is much lower than the former value in each system. Why? Explain it in the text.
- Figure 4. Show the TGA curves of OLA and a PHBH-OLA blend for comparison. They may give clearer analysis of the curves shown here.
- In what part of the composite do the authors think the compatibilizer OLA exist? Is it mixed with PHBH or phase-separated with it to settle in the PHBH-ASF interface?
Anyway, a deeper analysis is needed for characterization.
Author Response
Changes according Reviewer 2
This manuscript deals with the properties of PHBH-ASF composites as well as those added with oligolactic acid (OLA). The authors carefully conducted their experiments, and the data shown here may be reliable. I was however revealed that the properties of these composites are not so highly improved as expected probably because the interfacial interaction between the matrix polymer and fillers is weak. The control of ductility (toughness) and hardness (strength) is in a trade-off relation with the amount of fillers and plasticizer. Therefore, the present manuscript must be revised by considering the following points prior to its publication.
The title: “reactive extrusion oligomers” Why is this phrase necessary? Its explanation should be done somewhere in the text.
ANSWER
We agree with the reviewer. In order to justify the title, information on reactive extrusion has been added at the end of the introduction.
The information is shown below:
“Due to the lack of compatibility between the different elements, reactive extrusion (REX) has been proposed as a strategy to improve the properties of the mixtures. This process will improve the chemical bonding of the biopolymer chains to the surface of the lignocellulosic fillers by the action of reactive molecules with at least two functional sites.”
1 Materials. The molecular weight and unit composition of PHBH (i.e., 3HH unit ratio) must be shown. The degree of polymerization of OLA should also be shown here. These are fundamental characteristics of polymers. The size distribution of ASF must be shown. It can be calculated from Figure 1.
ANSWER
Thanks for this question. In response to the reviewer, the manufacturer has not provided us with that information for both PHBH and OLA. This works was focused on studying the potential of this PHBH/ASF composites compatibilized with OLA. Future works will focus on an in-depth characterization of the above-mentioned materials and their effect on processing by injection moulding. Regarding the size distribution, as it is quite irregular, we have calculated and added the average size from Figure 1 and added this information in the revised version.
Figure 2. No arrow is shown although it is explained in line 276.
ANSWER
We agree with this comment. The white arrows have been added to the image. In this way, the difference between the material gap can be seen better. In addition, the figure has been modified to show the rest of the materials and to see in greater depth the changes in morphology.
1 Mechanical properties……. As mentioned above, the present composites seem to have less improved mechanical properties even when OLA is added as a compatibilizer. In this regard, the authors are requested to show the target values of mechanical properties that are needed for their real application as WPC. Then, it should be discussed how well the present composites justify the indices or how highly balanced properties they have. Without this discussion, no one understands whether the present composites are really versatile or not.
ANSWER
In response to the reviewer, the incorporation of loads generally promotes a significant loss in mechanical properties of the WPCs, since there is poor interactions between the highly hydrophobic matrix and the highly hydrophilic lignocellulosic particles. Wood plastic composites are used in medium-to-low mechanical technical requirements. Therefore, WPCs represent an economic way to obtain cost-effective materials with balanced properties but being the wood-like appearance one of the most important issue. Technical requirements are dependent on the final application of the WPC. It is not the same for a fence, a decking, garden uses, furniture, and so on. Therefore, it is complicated to give a standard value or target to reach. In this work, the obtained materials are characterized by low tensile strength and slightly improved elongation at break. So that, these materials can be used in applications that require some flexibility and low mechanical resistance.
To improve the reader's understanding, a paragraph has been added to explain the balance obtained with this WPC on the mechanical properties.
“It is worthy to note that WPCs are widely used in applications that include fencing, garden objects, furniture, decking, and so on. The technical requirements will depend on the final application. AS per the mechanical results, these WPCs offer relatively low tensile strength and low elongation at break even without ASF filler. The addition of ASF up to 30 wt% and OLA as compatibilizer, gives interesting materials with a wood-like appearance but they cannot be used for medium technological applications as mechanical properties are low. Moreover, addition of 30 wt% ASF leads to a cost-effective material as PHBH matrix is still an expensive bacterial polyester.”
Figure 3 and Table 3. The heat of crystallization and heat of crystal fusion are not correspondent to each other. The latter value is much lower than the former value in each system. Why? Explain it in the text.
ANSWER
Following the recommendations of the reviewer, we have gone back to the raw DSC thermograms and we have completed the thermal analysis characterization. A third melting hump at 162 ºC has been also considered for calculating the melting enthalpy of PHBH (we have already added secondary literature to support PHBH melting process in three overlapped peaks). Now, the results we present are more coherent than previous.
|
Code |
Tg (ᵒC) |
Tcc (ᵒC) |
Tm1 (ᵒC) |
Tm2 (ᵒC) |
Tm3 (ᵒC) |
ΔHm1 (J g-1) |
ΔHcc2 (J g-1) |
ΔHm2 (J g-1) |
Xc1 (%) |
Xc2 (%) |
|
PHBH |
0.3±0.1 |
54.6±1.1 |
111.5±1.9 |
130.8±2.0 |
162.5±1.2 |
20.3±0.5 |
26.7±0.8 |
33.6±1.3 |
13.9±1.1 |
4.7±0.3 |
|
PHBH-10ASF |
-0.5±0.1 |
57.5±1.8 |
112.1±1.8 |
129.7±1.7 |
162.3±1.8 |
19.6±0.4 |
17.4±0.4 |
29.0±2.2 |
14.9±1.1 |
8.8±0.4 |
|
PHBH-20ASF |
-1.9±0.2 |
55.5±1.9 |
111.5±2.0 |
129.6±2.1 |
163.4±1.4 |
18.1±0.3 |
15.0±0.1 |
26.4±2.1 |
15.5±0.8 |
9.8±0.7 |
|
PHBH-30ASF |
-1.1±0.1 |
51.8±1.2 |
111.3±1.3 |
130.6±1.9 |
164.1±1.2 |
8.5±0.2 |
3.7±0.3 |
21.4±0.9 |
8.3±0.7 |
17.3±0.9 |
|
PHBH-30ASF/10OLA |
-5.2±0.2 |
56.3±1.4 |
107.6±2.1 |
127.0±1.8 |
159.4±1.6 |
8.2±0.1 |
11.4±0.1 |
26.12±1.3 |
8.8±0.8 |
14.7±0.8 |
|
PHBH-30ASF/20OLA |
-5.6±0.3 |
62.6±2.0 |
109.9±2.3 |
127.5±2.0 |
157.9±1.5 |
6.4±0.3 |
15.9±0.4 |
22.6±1.5 |
7.5±1.2 |
6.7±0.4 |
“The addition of a filler takes effect on the degree of crystallinity. During the first heating scan PHBH reaches a Xc value of 13.9%. This increases with the amount of ASF filler up to 15.5% with 20 wt% ASF which suggests that ASF (mainly crystalline cellulose fractions) acts as a nucleant agent [61]. Furthermore with the addition of 30 wt% ASF, Xc1 decreases to 8.3% due to the decrease of free volume necessary for nucleation of polymer, since high lignocellulosic filler content can hinder crystal formation [62]. Mechanical characterization shows no correlation between the degree of crystallinity, while Elastic Modulus increases up to 30 wt% ASF, the degree of crystallinity is saturated with only 20 wt%. The degree of crystallinity is different in the second heating scan because the recrystallization conditions are different. While on the first heating scan PHBH has been aged at 25 °C for 15 days, on the second heating scan the materials were recrystallized under controlled conditions above-mentioned. On the second heating scan, the same tendency is observed until 30 wt% ASF. The compatibilizing effect of OLA could negatively contribute increase the degree of crystallinity by reducing the gaps between the filler and the matrix as it is reported in FESEM analysis [63].”
Figure 4. Show the TGA curves of OLA and a PHBH-OLA blend for comparison. They may give clearer analysis of the curves shown here.
ANSWER
We agree with the reviewer. However, these TGAs were not performed previously and currently for security reasons (due to COVID-19) we cannot go to University. We think this is interesting but it is not a critical issue. This is our initial work with PHBH as it is not widely available. Therefore, we are planning new research works with PHBH which include the following:
- Effect of OLA and other plasticizers on PHBH properties.
- Effect of the heated mold during cooling after injection moulding.
- 3D- printing with PHBH for medical applications.
These are some examples of the panned works with this bacterial polyester. Therefore we plan to address the above-mentioned TGAs characterization in some of these future works.
In what part of the composite do the authors think the compatibilizer OLA exist? Is it mixed with PHBH or phase-separated with it to settle in the PHBH-ASF interface?
Anyway, a deeper analysis is needed for characterization.
ANSWER
By analyzing all the obtained results, it seems OLA is not fully miscible with PHBH since the decrease in the Tg is not significant as detected by DSC, DMTA and TMA techniques. It seems PHBH+OLA does not follow the Fox equation. Therefore, it could be possible to suggest that OLA locates between PHBH and lignocellulosic particles, and due to the carboxyl and hydroxyl terminal groups in OLA, it is possible the reaction of these groups with hydroxyl groups in cellulose and in terminal groups of the bacterial polyester, thus leading to a compatibilization effect, superior to a potential plasticization effect. To support this, Figure 2 has been redrawn containing FESEM images of PHBH-ASF/OLA composites, which give more clarity to that hypothesis. Composites without OLA show ASF particles with a small gap between them and the surrounding PHBH-rich matrix. Nevertheless, PHBH-ASF composites with OLA, do not show this gap; ASF particles seem to be completely embedded into the PHBH-rich matrix. This could corroborate the hypothesis that the main effect of OLA is compatibilization, instead of plasticization.

Round 2
Reviewer 1 Report
The authors significantly improve the manuscript by providing clear answers and additional information. I think the paper can be published. I have only few comments. I invite the authors to consider them before sending their final version.
1) p7: Elongation at break is still interpreted in terms of plasticization. I think it will be interesting to discuss the compatibilization.
2) p10: It is ambiguous in my opinion to write that ASF hinders the cold crystallization because it suggests that ASF slows down the crystallization kinetic. This is not necessary the case. The degree of crystallinity increases with the ASF percentage, meaning that crystallization occurs at higher extent during cooling. And consequently it occurs at lesser extent during the subsequent heating. Therefore, one can also think that ASF promotes crystallization. It is true that Xc1 is lower for ASF 30% but it is more difficult to compare the behaviour of materials being aged 15 days (by the way the authors propose a scenario involving free volume that seems credible to me).
To sum up, the enthalpy of cold crystallization depends on the thermal history (such as the previous cooling step). It has to be discussed cautiously, especially when samples are not all 100% amorphous.
3) I invite the authors to check again the typo before sending the final version as some slight mistakes may be present.
Line 112 requires
Line 175 was calculated
Line 408 be representative
Line 448 contribute to increase
Line 460 Reaching the endset,
Line 492 identical to neat PHBH
Author Response
May 8, 2020
Dear colleagues,
Thank you very much for your e-mail dated on May 6, 2020 regarding the manuscript entitled “Development and characterization of sustainable composites from bacterial polyester poly(3-hydroxybutyrate-co-3-hydroxyhexanoate) and almond
shell flour by reactive extrusion with oligomers of lactic acid” which was allocated with reference number polymers-784343.
We are sending the revised version of the manuscript that includes all suggestions and corrections proposed by the reviewers. In addition, an in depth checks of the grammar and spelling has been carried out and all detected mistakes have been corrected. Changes done to manuscript have been emphasized in yellow in order to facilitate their searching. We have worked in accordance with all reviewer comments so we consider that the version we are sending to you includes all necessary changes.
Sincerely,
PhD Student Juan Ivorra-Martinez
Institute of Materials Technology
Universitat Politècnica de València
Plaza Ferrándiz y Carbonell 1, 03801, Alcoy, Alicante (Spain)
e-mail: juaivar@doctor.upv.es
Changes according Reviewer 1
1) p7: Elongation at break is still interpreted in terms of plasticization. I think it will be interesting to discuss the compatibilization.
ANSWER
We thank the reviewer for this comment. Following the recommendations of the reviewer, in p7 and p9 additional comments are now incorporated to the text in order to comment the elongation at break in terms of compatibilization.
“Nevertheless, the addition of OLA leads to an improvement of the elongation at break due to a compatibilization between PHBH/ASF by the interaction of compatibilizer with terminal groups of PHBH and lignocellulosic particles. Also, a plasticization effect on the matrix can be expected by OLA acting as lubricant inside de polymer chain. Both effects were reported by Quiles-Carillo et al. with different biobased and petroleum-derived compatilibizers on PLA/ASF [38]. “
“The compatibilization effect reported in FESEM images (gap reduction) provided a more efficient load transfer between PHBH and ASF leading to an improvement of elongation at break as Quiles-Carrillo et al. reported [51]. “
2) p10: It is ambiguous in my opinion to write that ASF hinders the cold crystallization because it suggests that ASF slows down the crystallization kinetic. This is not necessary the case.
The degree of crystallinity increases with the ASF percentage, meaning that crystallization occurs at higher extent during cooling. And consequently it occurs at lesser extent during the subsequent heating. Therefore, one can also think that ASF promotes crystallization. It is true that Xc1 is lower for ASF 30% but it is more difficult to compare the behaviour of materials being aged 15 days (by the way the authors propose a scenario involving free volume that seems credible to me).
To sum up, the enthalpy of cold crystallization depends on the thermal history (such as the previous cooling step). It has to be discussed cautiously, especially when samples are not all 100% amorphous.
ANSWER
We agree with this comment. In order to remove the ambiguity, the affirmation has been removed. Moreover, just to clarify the explanation of the effect of ASF on the degree of crystallinity some modifications have been made in this revised version we are sending.
“On the first heating scan the polymer has been recrystallized at 25°C for 15 days. Consequently, during the first heating scan the cold crystallization peak did not appear which means that the polymer structure wasn’t able to form crystallites. Under this conditions PHBH reaches a Xc1 value of 13.9%. This increases with the amount of ASF filler up to 15.5% with 20 wt% ASF which suggests that ASF (mainly crystalline cellulose fractions) acts as a nucleant agent [62]. Furthermore with the addition of 30 wt% ASF, Xc1 decreases to 8.3% due to the decrease of free volume necessary for nucleation of polymer as Thomas et al. reported [63]. Mechanical characterization shows no correlation between the degree of crystallinity, while Elastic Modulus increases up to 30 wt% ASF, the degree of crystallinity is saturated with only 20 wt%. The second scan was performed after a controlled cooling process of 10 °C min-1, as a result in the second scan the polymer was able to form crystallites due to a cold crystallization process. Under this condition the degree of crystallinity could increase until 30 wt% ASF. The compatibilizing effect of OLA in both conditions decreased the degree of crystallinity by reducing the gaps between the filler and the matrix as it is reported in FESEM analysis and Gong et al. proposed [64].”
3) I invite the authors to check again the typo before sending the final version as some slight mistakes may be present.
Line 112 requires
Line 175 was calculated
Line 408 be representative
Line 448 contribute to increase
Line 460 Reaching the endset,
Line 492 identical to neat PHBH
ANSWER
Before sending the revised version (R1), we carefully checked grammar mistakes and typos, but as the reviewer has indicated, there are still some typos left in this version. Therefore, all identified typos by the reviewers and others that have arisen during double-checking, have been corrected.

Reviewer 2 Report
The authors have almost properly revised their manuscript by taking the referees' comments into consideration except the molecular characteristics of the polymers used.
Author Response
May 8, 2020
Dear colleagues,
Thank you very much for your e-mail dated on May 6, 2020 regarding the manuscript entitled “Development and characterization of sustainable composites from bacterial polyester poly(3-hydroxybutyrate-co-3-hydroxyhexanoate) and almond
shell flour by reactive extrusion with oligomers of lactic acid” which was allocated with reference number polymers-784343.
Sincerely,
PhD Student Juan Ivorra-Martinez
Institute of Materials Technology
Universitat Politècnica de València
Plaza Ferrándiz y Carbonell 1, 03801, Alcoy, Alicante (Spain)
e-mail: juaivar@doctor.upv.es
Reviewer 2
The authors have almost properly revised their manuscript by taking the referees' comments into consideration except the molecular characteristics of the polymers used.
ANSWER
We are very sorry for this. We have tried to get in touch with the suppliers of PHBH and OLA and they always refer us to the Technical DataSheet. We take this comment into account for further investigations because, it is a relevant information to include in the manuscript. As our investigations with PHBH and OLAs for different applications are ongoing, and once the access to the University and labs is allowed, we will obtain this information and will provide this in next published papers, related to these materials. In fact, right now we have an ongoing research on these OLAs to provide improved impact strength, chain extension and plasticization. On the other hand, we are running several works with PHBH as a material for 3D printing in medical applications. Therefore, we will obtain this information thorough characterization if the supplier does not give us that information.
